# BioMD: All-atom Generative Model for Biomolecular Dynamics Simulation

**Bin Feng**[1][*] **Jiying Zhang**[1,2][*] **Xinni Zhang**[1]**, Zijing Liu**[1][†]**, Yu Li**[1][†]

[1]International Digital Economy Academy, [2]Swiss Federal Technology Institute of Lausanne (EPFL)

## Abstract

Molecular dynamics (MD) simulations are essential tools in computational chemistry and drug discovery, offering crucial insights into dynamic molecular behavior. However, their utility is significantly limited by substantial computational costs, which severely restrict accessible timescales for many biologically relevant processes. Despite the encouraging performance of existing machine learning (ML) methods, they struggle to generate extended biomolecular system trajectories, primarily due to the lack of MD datasets and the large computational demands of modeling long historical trajectories. Here, we introduce BioMD, the first all-atom generative model to simulate long-timescale protein-ligand dynamics using a hierarchical framework of forecasting and interpolation. We demonstrate the effectiveness and versatility of BioMD on the DD-13M (ligand unbinding) and MISATO datasets. For both datasets, BioMD generates highly realistic conformations, showing high physical plausibility and low reconstruction errors. Besides, BioMD successfully generates ligand unbinding paths for 97.1% of the protein-ligand systems within ten attempts, demonstrating its ability to explore critical unbinding pathways. Collectively, these results establish BioMD as a tool for simulating complex biomolecular processes, offering broad applicability for computational chemistry and drug discovery.

## 1 Introduction

Molecular dynamics (MD) simulations have emerged as an indispensable tool in computational chemistry and drug discovery, offering insights into the dynamic behavior of biomolecular systems. Through numerical integration of Newton's equations of motion, MD simulations directly produce atomic trajectories that reveal the time evolution of molecular structures (Hollingsworth & Dror, 2018). These trajectories enable the exploration of conformational ensembles, optimization of small molecule structures, and identification of potential binding sites, significantly accelerating the design and development of novel therapeutics (Karplus & McCammon, 2002).

Despite their utility, traditional MD simulations face substantial computational limitations. Although algorithms like Particle Mesh Ewald (PME) have improved scaling to approximately $O(N \log N)$ (Darden et al., 1993), these calculations remain the most computationally demanding component (Dror et al., 2012; Adcock & McCammon, 2006). Furthermore, accurately resolving high-frequency atomic vibrations necessitates extremely small time steps (on the order of femtoseconds), severely limiting the accessible simulation timescales (Shaw et al., 2010; 2009). Exploring biologically relevant processes, which often span microseconds to milliseconds, remains computationally intensive, restricting the practical application of atomistic MD to obtain trajectories.

Recently, machine learning (ML) methods have emerged as computational alternatives to molecular dynamics (MD) simulations. Key advances include models for generating protein conformation ensembles (Lewis et al., 2025) and neural network potentials trained on quantum mechanical data (Wang et al., 2024a). For biomolecular systems, AlphaFold 3 (Abramson et al., 2024) has demonstrated promising accuracy in predicting protein–ligand interactions. Despite these achievements, generating full MD trajectories for complex protein–ligand systems using ML remains a major challenge. Existing approaches tend to fall into two categories: (i) methods that can generate protein conformation

---

[*]These authors contributed equally to this work.

[†]Corresponding authors: liuzijing@idea.edu.cn, liyu@idea.edu.cn

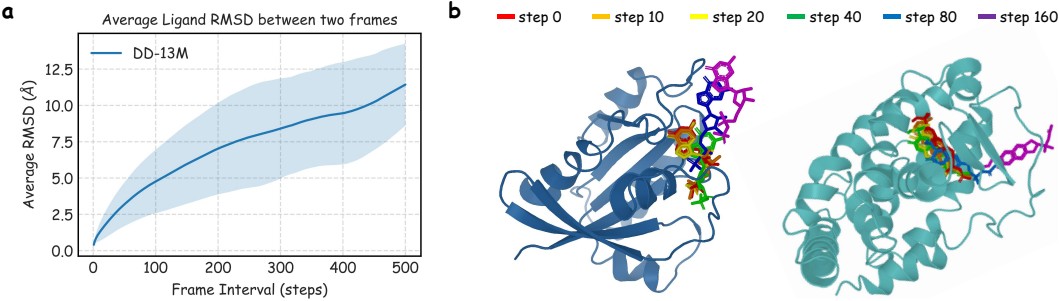

**Figure 1. Average Ligand RMSD between two frames.** (a) Line plot showing that the average ligand RMSD between two frames in the same trajectory increases with the frame interval. (b) Examples of ligand unbinding trajectories at time steps 0, 10, 20, 40, 80, and 160.

ensembles but cannot produce time-resolved trajectories Jing et al. (2024b); Wang et al. (2024b), or (ii) methods that attempt trajectory modeling but struggle to capture protein–ligand interactions. For example, NeuralMD (Liu et al., 2024) treats protein atoms as static and only models ligand dynamics, while MDGen (Jing et al., 2024c) is specifically designed for peptides and proteins and does not handle small-molecule ligands. This limitation arises from both the complexity of protein-ligand energy landscapes and the scarcity of high-quality trajectory data for training generative models.

To address these limitations, we propose BioMD, a hierarchical framework for generating all-atom biomolecular trajectories. Building upon the insight that short-timescale conformational changes exhibit little conformational change (**Figure 1**), BioMD decomposes long trajectory generation into two synergistic stages: forecasting of large-step conformations, followed by interpolation to refine intermediate steps. This strategy reduces sequence length by decoupling long-term evolution from local dynamics and helps manage the error accumulation problem for generating long trajectories. Crucially, BioMD unifies forecasting and interpolation within a conditional flow matching model, where we use the "noising-as-masking" methods following Diffusion Forcing (Chen et al., 2024) to our time-scale transformer. We apply independent noise to each frame, which enables flexible conditioning on partial trajectory segments, and we implement different tasks simply by using different masking schedules. Inspired by the success of AlphaFold 3, BioMD generates all-atom trajectories using a velocity network that adapts its core transformer architecture, while employing an SE(3)-equivariant graph transformer to encode the initial conformation as conditional embeddings.

To evaluate the effectiveness of BioMD, we conducted experiments on two datasets: MISATO (Siebenmorgen et al., 2024) and DD-13M (Li et al., 2025). Our results show that BioMD generates highly realistic conformations with promising physical stability, evidenced by low energy and reconstruction errors across both benchmarks. On the MISATO dataset, which focuses on ligand dynamics within the binding pocket, our model accurately captures the system's conformational flexibility, outperforming existing methods. For the more challenging task of ligand unbinding on the DD-13M dataset, BioMD successfully generates complete unbinding paths for up to 97.1% of the protein-ligand systems, demonstrating a robust ability to explore critical and long-timescale biomolecular pathways. Collectively, these results establish BioMD as a powerful and efficient tool for simulating complex biomolecular processes, offering broad applicability for computational chemistry and drug discovery.

## 2 RELATED WORKS

**Conformational Ensemble and Binding Pose Generation.** One major line of research uses ML to generate a biomolecule's conformational ensemble by modeling the equilibrium distribution of its dynamic structures. Early efforts like AlphaFold2 (Jumper et al., 2021) produce a set of diverse conformations primarily through multiple sequence alignment (MSA) subsampling and masking techniques (Stein & Mchaourab, 2022; del Alamo et al., 2022; Wayment-Steele et al., 2024). More advanced approaches now directly learn the conformational distribution from large-scale MD datasets using flow-based (Noé et al., 2019; Jing et al., 2024b) or diffusion-based (Wang et al., 2024b; Jing et al., 2023; Lu et al., 2024a; Zheng et al., 2024; Lu et al., 2025a) generative models. Models such as BioEmu (Lewis et al., 2025) can effectively generate diverse and physically plausible conformations, providing a powerful alternative to extensive MD sampling to understand a conformational space.

Beyond proteins, recent works have extended these generative frameworks to protein-ligand systems. DynamicBind (Lu et al., 2024b) and DynamicFlow (Zhou et al., 2025) leverage diffusion and flow matching models, respectively, to recover key protein-ligand binding poses while accounting for receptor flexibility. However, these methods are fundamentally time-agnostic; they can sample what conformations are possible (or the final binding states) but lack the temporal information to show the kinetic pathways between them.

**Trajectory Learning for MD Simulation.** To capture these kinetic pathways, a complementary research direction aims to generate full, time-ordered trajectories. Approaches like EquiJump (Costa et al., 2024) learn to sample future states based solely on the current conformation. To capture higher-order dependencies between the frames, MDGen (Jing et al., 2024c) models the joint distribution of entire trajectories via masked frame modeling. ConfRover (Shen et al., 2025) models these dependencies auto-regressively by conditioning each frame on its entire history through a causal transformer. While powerful, these methods are often specialized for protein-only dynamics. Conversely, methods that model protein-ligand interactions often introduce other simplifications. For instance, NeuralMD (Liu et al., 2024) treats the protein receptor as static. These simplifications limit their scope of accessible dynamics.

## 3 PRELIMINARIES

**Notations.** A complex $\mathcal{C}$ is composed of a protein $\mathcal{P}$ and a ligand $\ell$. The trajectory of a complex contains $T + 1$ frames of coordinates, denoted as $\mathbf{X}_T = \{\mathbf{x}_0, \mathbf{x}_1, \cdots \mathbf{x}_T\} \in \mathbb{R}^{(T+1) \times N \times 3}$, where $\mathbf{x}_t = [\mathbf{x}_t^{\mathcal{P}}, \mathbf{x}_t^{\ell}] \in \mathbb{R}^{N \times 3}$ represents the concatenation of protein coordinates $\mathbf{x}_t^{\mathcal{P}}$ and ligand coordinates $\mathbf{x}_t^{\ell}$ at time-step $t$, and $N$ is the number of atoms in the complex. The complex trajectory prediction task is defined as generating subsequent conformations (coordinates) of a complex trajectory given its initial conformation (i.e., the first frame).

**Molecular dynamics.** Molecular dynamics (MD) simulates the time evolution of a particle system under classical mechanics. It leverages numerical schemes such as Verlet integration (Verlet, 1967) or Langevin dynamics to generate trajectories approximating the Boltzmann distribution. In the simplest deterministic case with no friction or noise, each particle $i$ evolves according to $\mathrm{d}x_i = \frac{p_i}{m_i} \mathrm{d}t, \mathrm{d}p_i = -\nabla_{x_i} E(x) \mathrm{d}t$, where $p_i$ and $m_i$ are the momentum and mass, and $E(x)$ is the potential energy function. Metadynamics (Laio & Parrinello, 2002; Barducci et al., 2011; Li et al., 2025) extends MD by introducing a history-dependent bias potential $V(s, t)$, constructed over collective variables $s(x)$ as $V(s, t) = \sum_{t' < t} w \exp\left(-\frac{\|s(x(t)) - s(x(t'))\|^2}{2\sigma^2}\right)$, where Gaussians of height $w$ and width $\sigma$ are periodically added to discourage revisiting explored states. This bias fills free-energy wells and enhances sampling of rare events and transition pathways beyond the reach of standard MD.

**Flow matching based models.** Flow matching (FM) (Lipman et al., 2023) is an efficient and simulation-free method for training continuous normalizing flows (CNFs), a class of generative models based on ordinary differential equations (ODEs). In Euclidean space, CNFs define a transformation $\phi_\tau(\cdot) : \mathbb{R}^{N \times 3} \to \mathbb{R}^{N \times 3}$ via an ODE governed by a time-dependent velocity field $v_\tau$:

$$\frac{d}{d\tau} \phi_\tau(\mathbf{x}^0) = v_\tau(\phi_\tau(\mathbf{x}^0)), \quad \phi_0(\mathbf{x}^0) = \mathbf{x}^0, \quad \tau \in [0, 1], \tag{1}$$

where $\mathbf{x}^0$ is sampled from a simple distribution $p_0$, and $\phi_\tau$ evolves it over time $\tau \in [0, 1]$ to match the target distribution $p_1$ at $\tau = 1$. Since $v_\tau$ is unknown, FM learns $v_\tau$ by regressing the conditional flow $u(\phi_\tau(\mathbf{x}^0|\mathbf{x}^1)) = \frac{d}{d\tau} \phi_\tau(\mathbf{x}^0|\mathbf{x}^1)$, where $\phi_\tau(\mathbf{x}^0|\mathbf{x}^1)$ interpolates between $\mathbf{x}^0 \sim p_0$ and $\mathbf{x}^1 \sim p_1$. In our setting, each conformation $\mathbf{x}_t \in \mathbb{R}^{N \times 3}$ represents a frame in a complex trajectory, and FM is used to generate future frames from an initial structure.

## 4 BIOMD METHOD

### 4.1 A UNIFIED GENERATIVE FRAMEWORK VIA FLOW MATCHING

Our model capitalizes on a fundamental insight into molecular dynamics: conformational changes are typically subtle over short timescales but can involve significant global movements over longer

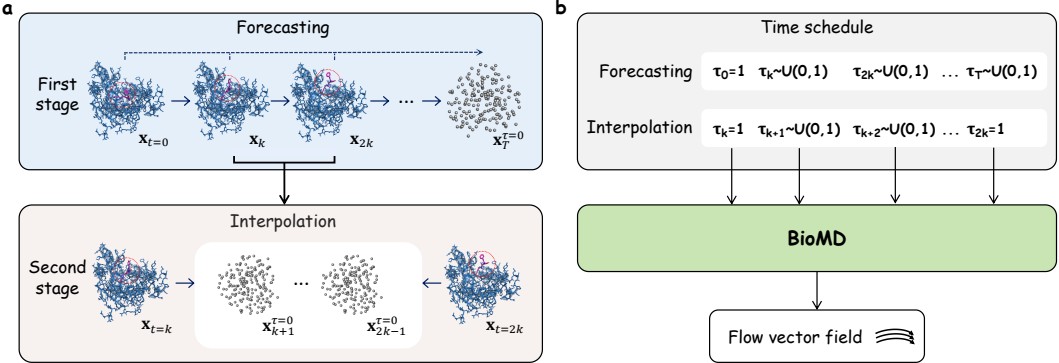

**Figure 2. Model framework.** (a) The hierarchical framework, showing the two-stage process of coarse-grained forecasting followed by fine-grained interpolation. (b) The time scheduling mechanism for forecasting and interpolation tasks, where known frames are noise-free ($\tau = 1$) and generated frames are noised ($\tau \in [0, 1]$).

timescales (**Figure 1**). This principle underpins our hierarchical prediction framework, which decomposes the generation of long trajectories into two principal stages: coarse-grained forecasting and fine-grained interpolation (**Figure 2**).

Notably, this entire framework is implemented within a single model architecture that processes the sequence of the whole trajectory at once. We adopt a "noise as mask" strategy, where the distinction between the two stages is made simply by varying the input masking patterns (Figure 2b). In this unified framework, each frame in an input sequence is independently perturbed by noise according to a time variable $\tau$. Known or conditioning frames are kept clean (equivalent to setting their corresponding $\tau = 1$, i.e., "unmasked"), while frames to be generated are initialized from pure noise (equivalent to $\tau = 0$, i.e., "masked") and then iteratively denoised.

Let a trajectory sequence be denoted by $\mathbf{X} = \{\mathbf{x}_{t_1}, \mathbf{x}_{t_2}, \ldots, \mathbf{x}_{t_L}\}$. During training, we sample a vector of independent time steps $\mathbf{T} = \{\tau_{t_1}, \tau_{t_2}, \ldots, \tau_{t_L}\}$, where each $\tau_{t_i} \sim U(0, 1)$. The sequence is then noised to $\mathbf{X}^{\mathbf{T}} = \{\mathbf{x}_{t_1}^{\tau}, \ldots, \mathbf{x}_{t_L}^{\tau}\}$, where each frame is an interpolation between the real coordinates and Gaussian noise $\boldsymbol{\epsilon}_i \sim \mathcal{N}(\mathbf{0}, \mathbf{I})$: $\mathbf{x}_{t_i}^{\tau} = \tau_{t_i} \mathbf{x}_{t_i} + (1 - \tau_{t_i}) \boldsymbol{\epsilon}_i$. The corresponding ground-truth velocity field for the sequence is $\mathbf{U}^{\mathbf{T}} = \{\mathbf{u}_{t_1}^{\tau}, \ldots, \mathbf{u}_{t_L}^{\tau}\}$, with $\mathbf{u}_{t_i}^{\tau} = (\mathbf{x}_{t_i} - \mathbf{x}_{t_i}^{\tau})/(1 - \tau_{t_i})$.

Our velocity model $u_\theta$ takes the entire noisy sequence and conditioning information to predict the velocities for all frames simultaneously. The training objective is a Mean Squared Error loss over the entire sequence:

$$\mathcal{L}_{\text{flow}} = \text{MSE}(u_\theta(\mathbf{X}^{\mathbf{T}}, \mathbf{Z}, \mathbf{T}), \mathbf{U}^{\mathbf{T}}). \tag{2}$$

where $\mathbf{Z}$ contains static information including the first frame coordinate $\mathbf{x}_0$, amino acid sequence $\mathbf{s}$, and ligand atom types $\mathbf{a}$. We explore two modeling approaches: BioMD-rel, which predicts coordinate changes relative to an anchor frame, and BioMD-abs, which predicts absolute atomic coordinates. For clarity, we focus on the absolute coordinate prediction task below.

## 4.2 HIERARCHICAL GENERATION WITH FORECASTING AND INTERPOLATION

The two stages of our hierarchical framework are realized simply by applying different masking schedules to our unified model during training and inference.

### 4.2.1 COARSE-GRAINED FORECASTING

The first stage generates a coarse-grained trajectory, constructed by sampling every $k = 10$ steps (empirically chosen, see ablation study in Appendix A.5) from the full trajectory, resulting in a sequence $\mathbf{X}_C = \{\mathbf{x}_0, \mathbf{x}_k, \mathbf{x}_{2k}, \ldots\}$. This task is framed as a forecasting problem where, given the initial frame $\mathbf{x}_0$, the model must generate all subsequent frames.

This is achieved by applying a specific masking schedule to our unified framework. During training, the time step for the initial frame is fixed at $\tau_0 = 1$ (making it a known, "unmasked" condition), while the time steps for all other frames $\{\tau_k, \tau_{2k}, \ldots\}$ are sampled independently from $U(0, 1)$. The model $u_\theta$ is trained to predict the velocities for all frames in the sequence, conditioned on the initial frame.

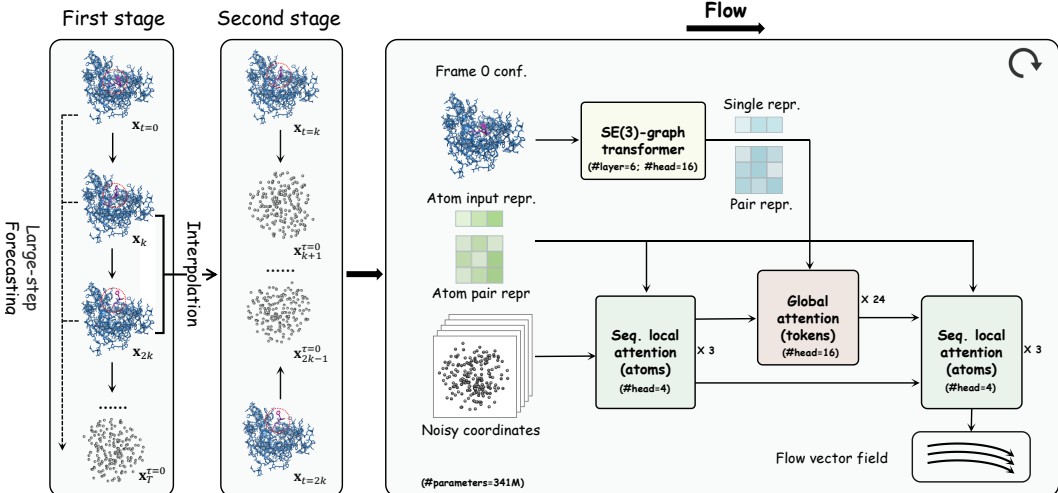

**Forecasting**: first frame is known, other frames begin with noise;
**Interpolation**: first frame and last frame are known, other frames begin with noise

**Figure 3. Detailed architecture of BioMD.** The model operates in two modes, **Forecasting** and **Interpolation**, set up by the hierarchical framework (left). The core velocity network (right) processes noisy coordinates, conditioned on features from an SE(3)-Graph Transformer. A local-global-local attention pathway generates the final flow vector field used for trajectory generation.

During inference, this setup supports multiple generation strategies:

- **All-at-once:** All future frames $\{\mathbf{x}_k, \mathbf{x}_{2k}, \dots\}$ are generated concurrently. We set $\tau_0 = 1$, initialize all other frames from noise (i.e., their $\tau$ values start at 0), and use an ODE solver like the Euler method to integrate all frames simultaneously to $\tau = 1$.

- **Auto-regressive (AR):** Frames are generated in sequential blocks of size $j$. To generate one such block, the model conditions on the previously generated history. This is controlled by the time variable $\tau$: the $\tau$ values for all frames in the history are set to be constant 1, making them clean, "unmasked" inputs. The $\tau$ values for all $j$ frames within the current target block are then jointly evolved from 0 to 1 by the ODE solver. This process simultaneously denoises all frames in the block, using the generated history as context. Once generated, this block is added to the history, and the process is repeated for the next block until the full trajectory is complete.

### 4.2.2 FINE-GRAINED INTERPOLATION

After obtaining the coarse-grained trajectory $\{\mathbf{x}_0, \mathbf{x}_k, \mathbf{x}_{2k}, \dots\}$, the second stage replenishes the intermediate frames. This is an interpolation task, where for each coarse interval, we generate the frames $\{\mathbf{x}_{ik+1}, \dots, \mathbf{x}_{(i+1)k-1}\}$ conditioned on the two "anchor" frames, $\mathbf{x}_{ik}$ and $\mathbf{x}_{(i+1)k}$.

This task uses the exact same velocity model $u_\theta$ and training framework, differing only in the data and masking schedule. The input sequence is now a fine-grained segment $\mathbf{X}_I = \{\mathbf{x}_{ik}, \mathbf{x}_{ik+1}, \dots, \mathbf{x}_{(i+1)k}\}$. During training, the anchor frames are designated as known by fixing their time steps $\tau_{ik} = 1$ and $\tau_{(i+1)k} = 1$. The time steps for all intermediate frames are sampled independently from $U(0, 1)$. The model learns to generate the intermediate trajectory conditioned on the start and end conformations.

During inference, this task is always performed in an "all-at-once" manner. The anchor frames $\mathbf{x}_{ik}$ and $\mathbf{x}_{(i+1)k}$ are provided as clean inputs (their $\tau = 1$), while all intermediate frames are initialized from noise (their $\tau = 0$). The model then simultaneously generates all $k - 1$ intermediate frames by integrating them to $\tau = 1$. This process is described by:

$$\hat{\mathbf{Y}}_{ik}^{\tau+\Delta\tau} = \hat{\mathbf{Y}}_{ik}^{\tau} + u_\theta(\hat{\mathbf{X}}_I^\mathbf{T}, \mathbf{Z}_{\text{seq}}, \mathbf{T}) \cdot \Delta\tau, \tag{3}$$

where $\mathbf{Y}_{ik}$ represents the block of intermediate frames, and the velocity predictions are extracted for only those frames. This hierarchical approach allows BioMD to efficiently generate long, physically plausible trajectories.

**Table 1. Results on the MISATO test set.** Comparison of all methods on physical stability (first six metrics) and conformational flexibility (last four metrics). Mean values on the test samples are reported.

| Method | Bond Geometry[a] | | Angle Geometry[a] | | Steric Clashes | | RMSF Correlation[b] | | RMSF Value[a,c] | |
|---|---|---|---|---|---|---|---|---|---|---|
| | MAE | MSE | MAE | MSE | Intra-Lig | Prot-Lig | Ligand | Protein | Ligand (1.211) | Protein (1.002) |
| Molecular Dynamics | .0377 | .0023 | .0575 | .0053 | 0 | 0 | - | - | - | - |
| DenoisingLD | $> 10^{10}$ | $> 10^{27}$ | .1018 | .0431 | .0160 | .0295 | -0.0290 | - | $> 10^{12}$ | - |
| GNNMD | .2123 | .1032 | .2115 | .1072 | .3626 | .0028 | -0.0103 | - | .2165 | - |
| NeuralMD-ODE | .0483 | .0076 | .0605 | .0086 | .0114 | .0578 | .3405 | - | .3220 | - |
| NeuralMD-SDE | .0483 | .0076 | **.0604** | .0086 | .0114 | .0578 | .3405 | - | .3220 | - |
| VerletMD | 19.73 | 1050 | .5847 | .5482 | .1983 | 3.111 | .3356 | - | .3226 | - |
| BioMD-rel | **.0395** | **.0026** | .0655 | **.0075** | **.0003** | **.0006** | **.4861** | .5945 | .5369 | .5177 |
| BioMD-abs | .0495 | .0155 | .0709 | .0097 | .0019 | .0023 | .4789 | **.6854** | **.7023** | **.6242** |

[a] Bond geometry (bond length) and RMSF values are in angstroms (Å). Angle geometry (bond angle) is in radians.
[b] RMSF Correlation is reported using the Pearson correlation coefficient.
[c] RMSF values for reference trajectories are given in parentheses. Values closer to those of the reference indicate better results.

## 4.3 VELOCITY MODEL ARCHITECTURE

BioMD is a generative model that operates directly on all-atom Cartesian coordinates. In contrast to approaches that rely on internal coordinates such as coarse-grained backbones and torsion angles, our method directly models all atoms, enabling it to capture subtle structural variations that are critical for realistic biomolecular dynamics. The effectiveness of this all-atom modeling strategy has been demonstrated by state-of-the-art biomolecular structure models like AlphaFold3 (Abramson et al., 2024). Notably, our unified model architecture is capable of performing both the forecasting and interpolation tasks (subsec. 4.2.1 and 4.2.2) within the same framework.

Our velocity model architecture is specifically tailored for generating trajectories from a single initial structure (**Figure 3**). The model first employs an SE(3) Graph Transformer to encode the initial conformation, creating rich single and pair representations. Subsequently, our core generative module, the `FlowTrajectoryTransformer` (**Algorithm 6**), operates on the entire trajectory sequence. To effectively capture complex biomolecular dynamics, each block of this transformer incorporates two primary attention mechanisms: `AttentionPairBias` is responsible for modeling intra-frame spatial interactions, while `TemporalAttention` specifically addresses inter-frame temporal dependencies by focusing on the same atom or token across different time steps. By stacking these two attention mechanisms, the model can simultaneously process spatial and temporal information, which is crucial for accurate trajectory prediction.

## 4.4 AUXILIARY LOSSES

In addition to the primary flow-matching objective, we incorporate several auxiliary losses to improve the physical plausibility of the generated structures. These losses are applied to the final predicted coordinates, which are obtained using the model's output velocity field.

- **Ligand Bond Loss:** To preserve the ligand's local structure, we introduce a bond loss following AlphaFold 3 (Abramson et al., 2024). For each bonded atom pair in the ligand, we compute the mean squared error between the predicted inter-atomic distance and its ground-truth value, ensuring that the generated ligand structure maintains correct bond lengths.

- **Collision Loss:** To ensure physical plausibility and prevent steric clashes, we implement a collision loss that applies a squared penalty to non-bonded atom pairs that are unrealistically close. This loss operates on both protein-ligand and intra-ligand interactions, and penalizes inter-atomic distance that falls below a predefined threshold.

- **Ligand Geometric Center Loss:** To penalize unrealistic rigid-body movements of ligands, we define a geometric center loss. This loss calculates the mean squared error between the geometric center of the predicted ligand atoms and that of the ground-truth ligand atoms, penalizing large and unrealistic movements of the entire molecule.

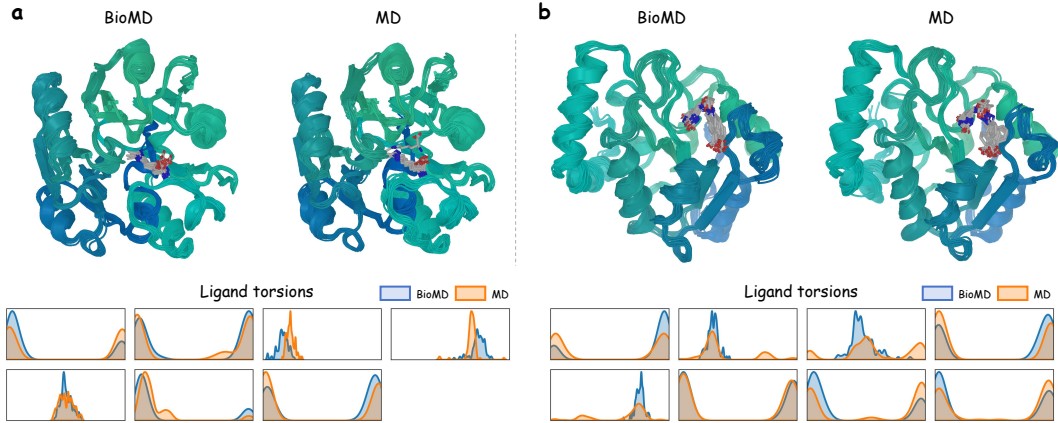

**Figure 4. Conformation ensemble on the MISATO test set.** A comparison of the distributions of conformations and ligand torsion angles generated by BioMD and MD simulation for 6DGE (**a**) and 3FCF (**b**).

## 5 EXPERIMENTS

We evaluate BioMD on three MD trajectory datasets: the MISATO Dataset (Siebenmorgen et al., 2024), which comprises protein-ligand interaction trajectories focusing on ligand movement within the protein binding pocket; the DD-13M Dataset (Li et al., 2025), which contains trajectories of ligand unbinding from protein binding pockets and ultimately reaching the protein surface; the ATLAS dataset (Vander Meersche et al., 2024), which contains 100 ns simulations for 1390 protein chains. Examples of predicted trajectories can be obtained from Zenodo. [1]

To comprehensively evaluate our model's performance in generating all-atom biomolecular trajectories, we first evaluate the physical stability of the generated structures for both MISATO and DD-13M. To enable direct comparison with methods that don't consider protein flexibility, physical stability metrics are computed for all heavy atoms of ligands. For the MISATO dataset, given that this dataset provides conformational ensembles, we further evaluate our model's ability to predict the conformational flexibility of both proteins and ligands. For the DD-13M ligand unbinding dataset, we also included several metrics to assess the accuracy of the predicted unbinding pathways. For experiments on the Atlas dataset, we evaluate all methods using metrics provided by Jing et al. (2024a). In this paper, we compare BioMD with several established ML methods, including methods developed for protein-ligand MD (e.g. NeuralMD (Liu et al., 2024)) and methods developed for protein conformation sampling (e.g. ESMFLow Jing et al. (2024a)). We also include a Static model as a baseline, where the initial conformation of the system is held constant throughout the entire trajectory. For the ATLAS dataset, results of Str2Str (Lu et al., 2024a) are obtained using the released code from the official repositories with $T_\delta$ set to 0.10 and 0.15 following their instructions, and evaluation results on other methods are provided by Lu et al. (2025b).

### 5.1 RESULTS ON MISATO

To evaluate BioMD's ability to generate realistic protein-ligand interaction trajectories, we first conduct experiments on the MISATO dataset, which focuses on ligand dynamics within the protein binding pocket. MISATO comprises nearly 20,000 protein-ligand interaction trajectories, each containing 100 frames sampled from an 8 ns MD simulation. We evaluate all methods on the MISATO dataset using 1,031 targets, each with a protein sequence length $\leq 800$ and a ligand size of $\leq$ 100 heavy atoms. As shown in **Table 1** and **Figure 7**, both variants of our model, BioMD-rel and BioMD-abs, produce trajectories with promising physical stability. The bond and angle geometry errors closely approach the values of the static input structure, and the steric clash scores are orders of magnitude lower than all competing models. Further validation on relaxation consistency (**Figure 8**) and pocket-ligand interactions (**Figures 9, 10**) confirms the effectiveness of BioMD to generate physically plausible structures.

---

[1]https://doi.org/10.5281/zenodo.16979768

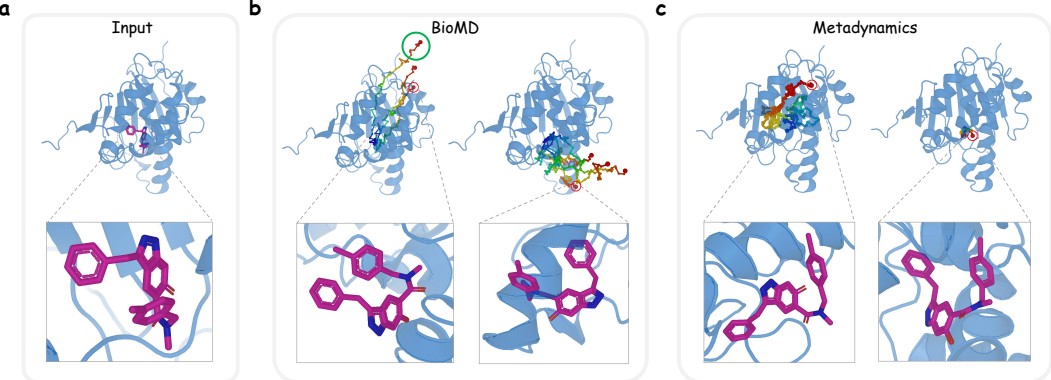

**Figure 5. Ligand unbinding path on 6EY8.** (a) The input conformation. (b) The unbinding pathways generated by BioMD, with an unbinding path RMSD of 0.24 Å(Metadynamics=0.12 Å). The novel pathway discovered by BioMD is highlighted in green. (c) The reference unbinding pathways obtained using metadynamics simulations.

**Table 2. Results on the DD-13M test set.** Comparison of methods on physical stability (first six metrics), ligand unbinding path reconstruction metric (Unbinding Path RMSD), and ligand unbinding success rates. Mean values on the test samples are reported.

| Method | Bond Geometry[a] | | Angle Geometry[a] | | Steric Clashes | | Unbinding Path[a] | Unbinding Success | | |
|---|---|---|---|---|---|---|---|---|---|---|
| | MAE | MSE | MAE | MSE | Intra-Lig | Prot-Lig | RMSD | @1 | @5 | @10 |
| Static | - | - | - | - | .2778 | 0 | .6504 | 0 | 0 | 0 |
| Metadynamics[b] | .0246 | .0012 | .0452 | .0030 | .2777 | 0 | .4217 | - | - | - |
| BioMD-rel | **.0308** | **.0018** | .0606 | .0077 | .2943 | .0004 | .6845 | .0029 | .0147 | .0294 |
| BioMD-abs | .0369 | .0026 | **.0545** | **.0061** | **.2941** | **.0003** | .6802 | .0176 | .0440 | .0588 |
| BioMD-rel (AR-5) | .0580 | .0100 | .0918 | .0184 | .4021 | .6375 | .7055 | **.7088** | **.9295** | **.9706** |
| BioMD-abs (AR-5) | .0728 | .0111 | .0802 | .0132 | .2943 | .0009 | **.5645** | .5676 | .7419 | .7941 |

[a] Bond geometry (bond length) and unbinding path RMSD values are in angstroms (Å), and angle geometry (bond angle) is in radians.
[b] The metadynamics trajectory serves as the lower-bound. The metrics are calculated among trajectories of multiple repeating simulations.

In terms of conformational flexibility, BioMD demonstrates a superior ability to capture the system's dynamic behavior. We measure Pearson's correlation between the Root Mean Square Fluctuation (RMSF) of our generated trajectories and the reference MD trajectories. BioMD achieves the highest correlation score for ligand atoms, outperforming NeuralMD by 42.8%. Besides, BioMD achieves the correlation score of 0.685 for protein atoms, while other comparing methods fail to simulate protein conformation changes. Visual analysis in **Figures 4, 11** further corroborates these findings, showing that BioMD's predicted atomic fluctuations closely trace the ground truth profiles and that the generated conformational ensemble is qualitatively similar to that of a traditional MD simulation. Collectively, these results indicate that BioMD can accurately simulate the flexibility of the entire protein-ligand complex.

## 5.2 RESULTS ON DD-13M

We further evaluate BioMD on the more challenging task of ligand unbinding using the DD-13M dataset, which comprises 26,612 dissociation trajectories across 565 complexes, each with an average of 480 frames. 36 complexes were held out as a test set for evaluation, while the remaining were used for training. As DD-13M is generated via metadynamics, the task focuses on reproducing sampling pathways, which do not necessarily represent true thermodynamic or kinetic behavior. A key advantage of our architecture is its flexibility in supporting multiple generation strategies. A concurrent denoising of all future frames, as used on MISATO, results in minimal ligand movement because the model lacks historical guidance and averages over many potential paths. To overcome this, we generate the trajectory auto-regressively, which breaks the long-range prediction into steps and uses previously generated frames to help predict subsequent ones.

The results, summarized in **Table 2**, highlight the effectiveness of this auto-regressive strategy. While maintaining high physical stability, the BioMD-abs (AR-5) model significantly improved path accuracy, reducing the Unbinding Path RMSD to 0.5645 Å and achieving a high unbinding correct rate

**Table 3. Results on the ATLAS test set.** Performance comparison of methods for modeling protein dynamics across different metrics. Bold values indicate the best performance for each metric.

| | | ESMFLow-MD | | | | | | | |
| | Metrics | Full | Distilled | ConfDiff | BioEmu | Str2Str | MDGen | EBA | BioMD |
|---|---|---|---|---|---|---|---|---|---|
| Predicting flexibility | Pairwise RMSD $r$ ↑ | 0.19 | 0.19 | 0.59 | 0.46 | 0.23 | 0.48 | 0.62 | **0.70** |
| | Global RMSF $r$ ↑ | 0.31 | 0.33 | 0.67 | 0.57 | 0.38 | 0.50 | 0.71 | **0.76** |
| | Per-target RMSF $r$ ↑ | 0.76 | 0.74 | 0.85 | 0.71 | 0.57 | 0.71 | 0.90 | **0.91** |
| Distributional accuracy | Root mean $W_2$-dist. ↓ | 3.60 | 4.23 | 2.76 | 4.32 | 4.05 | 2.69 | 2.43 | **2.18** |
| | ↪ Trans. contrib. ↓ | 3.13 | 3.75 | 2.23 | 4.04 | 3.73 | - | 2.03 | **1.89** |
| | ↪ Var. contrib. ↓ | 1.74 | 1.90 | 1.40 | 1.77 | 1.43 | - | 1.20 | **1.10** |
| | MD PCA $W_2$-dist. ↓ | 1.51 | 1.87 | 1.44 | 1.97 | 2.04 | 1.89 | **1.19** | 1.24 |
| | Joint PCA $W_2$-dist. ↓ | 3.19 | 3.79 | 2.25 | 3.98 | 3.55 | - | 2.04 | **1.82** |
| | % PC-sim > 0.5 ↑ | 26 | 33 | 35 | **51** | 12 | - | 44 | 46 |
| Ensemble observables | Weak contacts $J$ ↑ | 0.55 | 0.48 | 0.59 | 0.33 | 0.43 | 0.51 | **0.65** | 0.61 |
| | Transient contacts $J$ ↑ | 0.34 | 0.30 | 0.36 | - | - | - | 0.41 | **0.46** |
| | Exposed residue $J$ ↑ | 0.49 | 0.43 | 0.50 | - | - | 0.29 | 0.70 | **0.70** |
| | Exposed MI matrix $\rho$ ↑ | 0.20 | 0.16 | 0.24 | 0.07 | 0.21 | - | **0.36** | 0.34 |

(**Table 9**). Most importantly, the AR strategy enabled the successful generation of complete unbinding events. The BioMD-rel (AR-5) model achieved a remarkable unbinding success rate, identifying a valid path in 70.9% of cases with a single attempt (@1), increasing to 97.1% with ten attempts (@10). This demonstrates BioMD's reliability in exploring critical biomolecular pathways.

On the qualitative analysis for the 6EY8 system (**Figure 5**), our model not only reproduced the two distinct unbinding pathways found by metadynamics simulations with high fidelity but also discovered a novel third pathway, highlighting the exploratory power of our generative approach. Furthermore, BioMD achieves this with remarkable computational efficiency. While metadynamics required 2654 steps (approx. 1 hour on a single GPU) to find the first path, our model generated a complete path in under 10 seconds using just 50 coarse-grained steps. Additionally, in an out-of-distribution (OOD) case with low sequence similarity (**Figure 12**), BioMD successfully covers multiple escape clusters with an unbinding path RMSD of 0.77 Å (Metadynamics = 0.65 Å).

## 5.3 RESULT ON ATLAS

We finally used the Atlas dataset (Vander Meersche et al., 2024) to evaluate the model's ability to capture long-timescale protein dynamics with BioMD-rel. The Atlas dataset comprises 1,390 single-chain targets, each associated with three independent 100 ns Molecular Dynamics (MD) trajectories. We evaluated its performance on the Atlas test set, which comprises 82 targets. For each target, we sampled three 100 ns trajectories, where the initial frame for each trajectory was randomly selected from the corresponding MD reference trajectories. From the resulting 300 ns of generated trajectories, we extracted 250 frames for evaluation. As shown in **Table 3**, BioMD achieves state-of-the-art performance on 9 out of 13 metrics, demonstrating its superiority in capturing both structural flexibility and distributional accuracy. Compared to MDGen, which shares the same input setting (sequence + initial frame), BioMD exhibits substantial improvements across all metrics. Notably, we observe a 52% increase in the Global RMSF correlation coefficient ($r$), highlighting the effectiveness of our all-atom architecture in modeling residue-level fluctuations. Furthermore, BioMD surpasses EBA (Lu et al., 2025a), the leading sequence-based method, on the majority of distributional metrics. This confirms that conditioning on the initial structure provides critical geometric guidance for producing high-fidelity and dynamically relevant ensembles.

Beyond statistical metrics, BioMD successfully reproduces functional conformational transitions. A prime example is the domain motion of Adenylate Kinase (**Figure 6**). Starting from the closed state (1AKE, 0 ns), BioMD simulates the opening process, reaching a conformation at 23 ns that closely aligns with the experimental open state (4AKE) with an RMSD of 1.631 Å. This indicates that our model can capture major protein conformational transitions that are biologically significant.

To further analyze the exploration capability, we projected the generated conformations onto a 2D space using Time-lagged Independent Component Analysis (TICA). As shown in **Figure 13**, BioMD explores the conformational space more sufficiently than short-timescale MD simulations (1 ns and 10 ns), capturing a diversity closer to long MD trajectories (100 ns). Remarkably, BioMD generates a full 100 ns trajectory in approximately 56 seconds, offering orders-of-magnitude acceleration over traditional MD simulations.

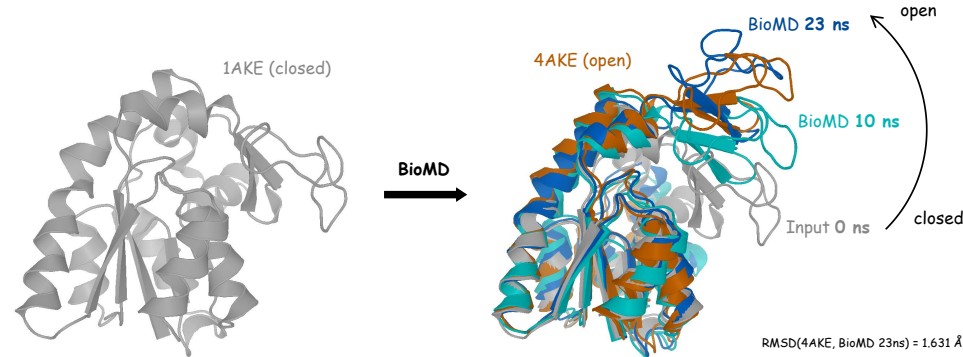

**Figure 6. Case study on the adenylate kinase**. BioMD simulates the domain motion of the adenylate kinase from close (0 ns) to open (23 ns). The test protein is not included in the training set.

## 5.4 ANALYSIS

The success of the auto-regressive (AR) strategy in modeling long-range dynamics simultaneously exposes a fundamental challenge in generative trajectory modeling: the error accumulation problem. As shown in **Table 1, 2 and Figure 7**, while the non-AR models produce local geometries with errors comparable to the metadynamics reference, the AR models exhibit a notable increase in error. However, thanks to our hierarchical framework, these errors remain manageable. The bond and angle MAEs for our AR models remain below 0.1 Å and 0.1 radians, respectively—a threshold well within the range of thermal fluctuations for molecular systems. These geometrical errors can be readily corrected via a simple local refinement step with minor structural deviations ($< 0.1$ Å), similar to the relaxation procedure used in AlphaFold. In contrast, non-hierarchical approaches are trapped between two failure modes: large AR steps yield nearly static trajectories, while small AR steps cause significant error accumulation that results in physically unrealistic structures.

Our results also reveal a distinct trade-off between predicting relative coordinate changes (BioMD-rel) and absolute coordinates (BioMD-abs). The absolute coordinate prediction method (BioMD-abs) demonstrates a superior grasp of the global conformational landscape, evidenced by its higher protein RMSF correlation on MISATO and a more accurate centroid path RMSD on DD-13M, making it the preferred choice for tasks requiring the precise reproduction of specific dynamic pathways. In contrast, the relative coordinate prediction method (BioMD-rel) excels at encouraging more exploratory behavior while preserving local chemical fidelity. Its strength is highlighted by the significantly higher unbinding success rate on DD-13M, which makes it more effective for applications focused on sampling large-scale conformational changes and discovering novel dynamic events. This functional duality means BioMD can be flexibly adapted to the specific goals of a simulation, whether the priority is accuracy in reproducing known dynamics or exploration to discover new ones.

## 6 CONCLUSION

In this work, we introduce BioMD, a novel all-atom generative model that overcomes the computational limitations of traditional molecular dynamics to simulate long-timescale biomolecular events. Our hierarchical framework, which synergistically combines coarse-grained forecasting with fine-grained interpolation, effectively mitigates error accumulation and enables the generation of physically realistic trajectories. We demonstrated BioMD's capabilities on two challenging datasets, showing it can produce stable conformations that accurately capture protein-ligand flexibility on the MISATO dataset and successfully generate complete ligand unbinding pathways for up to 97.1% of systems on the DD-13M dataset. Notably, BioMD achieves this with remarkable computational efficiency, identifying unbinding paths in seconds compared to the hours required by traditional methods like metadynamics. By offering distinct modes optimized for either accurate pathway reproduction or broad exploratory sampling, BioMD provides a powerful, flexible, and efficient tool poised to accelerate research in computational chemistry and drug discovery. Nevertheless, BioMD's ability to generalize to substantially longer trajectories ($\mu$s or ms) or rare events beyond the training distribution remains limited, and we highlight this as an important direction for future work. [2]

---

[2]This paper is written with assistance from large language models (LLM) for proofreading and polishing.

ACKNOWLEDGMENTS

This project was supported by Shenzhen Hetao Shenzhen-Hong Kong Science and Technology Innovation Cooperation Zone, under Grant No. HTHZQSWS-KCCYB-2023052.

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

## A  Technical Appendices and Supplementary Material

### A.1  Detailed Model Architecture

**Hierarchical Generation Framework.**    As illustrated in **Figure 3**, BioMD employs a hierarchical framework to perform both coarse-grained forecasting and fine-grained interpolation within a unified model. The specific task is controlled by applying noise selectively. For **Forecasting**, the initial frame $\mathbf{x}_0$ is provided without noise, while all subsequent frames are initialized from a standard Gaussian distribution. For **Interpolation**, two anchor frames (e.g., $\mathbf{x}_k$ and $\mathbf{x}_{2k}$) are kept clean, while the intermediate frames are initialized from noise. The model's objective is to denoise the masked frames conditioned on the known ones.

**Input Representation and Conditioning.**    The core of the model is the `FlowModule` (**Algorithm 4**), which processes three primary inputs. The main dynamic input is the set of Noisy Coordinates ($\{\vec{\mathbf{x}}_l^{\text{noisy}}\}$), representing the current state of the trajectory. To provide structural context, the initial conformation (Frame 0 conf.) is processed by an SE(3)-Graph Transformer, as detailed in the main inference loop (**Algorithm 1**). This produces static Single ($\{\mathbf{s}_i^{\text{trunk}}\}$) and Pair ($\{\mathbf{z}_{ij}^{\text{trunk}}\}$) representations. These representations, along with other atom features, are processed by the `FlowConditioning` module (**Algorithm 5**) to generate the final conditioning signals.

**Spatial-Temporal Attention Pathway.**    The `FlowModule` uses a local-global-local attention pathway to predict the velocity field. First, the noisy coordinates and conditioning features are passed to an `AtomAttentionEncoder`, which models local atomic environments. The resulting representations are aggregated into tokens and fed into the central `FlowTrajectoryTransformer` (**Algorithm 6**). This module integrates spatial and temporal information using two key mechanisms: `AttentionPairBias` resolves intra-frame spatial relationships, while `TemporalAttention` captures inter-frame dynamics. The globally-aware token representations are then broadcast back to the atomic level, where an `AtomAttentionDecoder` computes the final per-atom updates.

**TemporalAttention Module.**    The `TemporalAttention` module is a key component of our architecture, designed to capture dynamic dependencies across the time dimension of the trajectory. As shown in **Algorithm 7**, it operates on the single representation, $c_s$, by treating the time axis as the sequence length for the attention mechanism. This allows the model to integrate information from all frames of the trajectory for each residue or atom.

**AttentionPairBias Module.**    The `AttentionPairBias` module (**Algorithm 8**) originally comes from AlphaFold. It projects the pair representation $c_z$ into a bias term. This term is added directly to the attention logits before the softmax operation, effectively steering the attention mechanism to focus on spatially relevant residue pairs.

**Velocity Field Prediction and Trajectory Generation.**    The output of the `FlowModule` is the Flow vector field ($\{\vec{\mathbf{u}}_l\}$), which represents the predicted velocity for each atom. During training (**Algorithm 2**), the model is optimized via a mean squared error loss between the predicted velocity and the true velocity. During inference (**Algorithm 3**), this vector field is used in an Euler integration step, $\vec{\mathbf{x}}_l^{\tau+1} \leftarrow \vec{\mathbf{x}}_l^{\tau} + dt \cdot \vec{\mathbf{u}}_l^{\tau}$, to iteratively update the coordinates from a noisy state to a fina trajectory.

### A.2  Auxiliary losses

After we get the estimated vector field $\mathbf{u}_\theta$, we can get the predicted structure coordinates via

$$\hat{\mathbf{x}}_t^1 = \hat{\mathbf{x}}_t^\tau + \mathbf{u}_\theta(1 - \tau), \tag{4}$$

and then we get the predicted protein and ligand structure $[\hat{\mathbf{x}}_t^{\mathcal{P}}, \hat{\mathbf{x}}_t^{\ell}] = \hat{\mathbf{x}}_t^1$.

**Ligand geometric center loss.**    To stabilize the global placement of the ligand and prevent spurious rigid translations, we align the predicted and reference geometric centers of ligand atoms. Let $\mathbf{x}_t^{\ell} = \{\mathbf{x}_t^{\ell,i}\}_{i=1}^{N_\ell}$ and $\hat{\mathbf{x}}_t^{\ell} = \{\hat{\mathbf{x}}_t^{\ell,i}\}_{i=1}^{N_\ell}$ denote ground-truth and predicted ligand coordinates at step $t$.

The geometric center is

$$C(\mathbf{x}_t^\ell) = \frac{1}{N_\ell} \sum_{i=1}^{N_\ell} \mathbf{x}_t^{\ell,i}, \qquad C(\hat{\mathbf{x}}_t^{\ell,i}) = \frac{1}{N_\ell} \sum_{i=1}^{N_\ell} \hat{\mathbf{x}}_t^{\ell,i},$$

and the loss is the mean-squared discrepancy

$$\mathcal{L}_{\text{center}} = \left\| C(\hat{\mathbf{x}}_t^\ell) - C(\mathbf{x}_t^\ell) \right\|_2^2.$$

This term softly anchors the ligand's global position while remaining agnostic to its internal geometry.

**Collision loss.** To penalize steric clashes, we define a collision loss between protein–ligand atoms and within ligand atoms. Let $\mathbf{x}_t^\ell$ and $\mathbf{x}_t^{\mathcal{P}}$ denote ligand and protein atom coordinates at step $t$, and $\hat{\mathbf{x}}_t^\ell$, $\hat{\mathbf{x}}_t^{\mathcal{P}}$ their predictions. We compute predicted distances

$$d_{ij}^{PL} = \|\hat{\mathbf{x}}_t^{\mathcal{P},i} - \hat{\mathbf{x}}_t^{\ell,j}\|_2, \quad d_{ij}^L = \|\hat{\mathbf{x}}_t^{\ell,i} - \hat{\mathbf{x}}_t^{\ell,j}\|_2,$$

and corresponding ground-truth minimal distances

$$d_{ij}^{PL,gt} = \min_t \|\mathbf{x}_t^{\mathcal{P},i} - \mathbf{x}_t^{\ell,j}\|_2, \quad d_{ij}^{LL,gt} = \min_t \|\mathbf{x}_t^{\ell,i} - \mathbf{x}_t^{\ell,j}\|_2.$$

Protein–ligand and ligand–ligand thresholds are set as

$$\zeta_{ij}^{PL} = \min\left(0.9\, d_{ij}^{PL,gt},\ \zeta_{pl}\right) \quad \zeta_{ij}^{LL} = \min\left(0.9\, d_{ij}^{LL,gt},\ \zeta_{ll}\right),$$

where $\zeta_{pl} = 3.0$ Å and $\zeta_{ll} = 2.0$ Å.

The collision loss is then defined as

$$\mathcal{L}_{\text{collision}} = \sum_{i,j} \mathbf{1}\left(d_{ij}^{PL} < \zeta_{ij}^{PL}\right) (\zeta_{ij}^{PL} - d_{ij}^{PL})^2 + \sum_{i \neq j} \mathbf{1}\left(d_{ij}^{LL} < \zeta_{ij}^{LL}\right) (1 - b_{ij}) (\zeta_{ij}^{LL} - d_{ij}^{LL})^2,$$

where $\mathbf{1}(\cdot)$ represents the indicator function and $b_{ij}$ is the ligand bond mask to exclude bonded pairs.

**Ligand bond loss.** To preserve ligand bond lengths, we penalize deviations between predicted and ground-truth bonded atom distances. Let $\mathcal{B}$ denote the set of bonded atom pairs according to the ligand bond mask. For each bond $(i,j) \in \mathcal{B}$, we compute the predicted and ground-truth distances

$$d_{ij}^\ell = \|\hat{\mathbf{x}}_t^{\ell,i} - \hat{\mathbf{x}}_t^{\ell,j}\|_2, \quad d_{ij}^{\ell,gt} = \|\mathbf{x}_t^{\ell,i} - \mathbf{x}_t^{\ell,j}\|_2.$$

The bond loss is then defined as the mean squared deviation:

$$\mathcal{L}_{\text{bond}} = \frac{1}{|\mathcal{B}|} \sum_{(i,j) \in \mathcal{B}} \left(d_{ij}^\ell - d_{ij}^{\ell,gt}\right)^2.$$

**Geometric constraint loss.** We combine the above terms into a single geometric regularizer

$$\mathcal{L}_{\text{geom}} = \lambda_{\text{col}}\, \mathcal{L}_{\text{collision}} + \lambda_{\text{bond}}\, \mathcal{L}_{\text{bond}} + \lambda_{\text{ctr}}\, \mathcal{L}_{\text{center}},$$

where $\lambda_{\text{col}}, \lambda_{\text{bond}}, \lambda_{\text{ctr}} > 0$ balance steric clash avoidance, bond-length preservation, and global ligand anchoring, respectively.

## A.3 EVALUATION METRICS

### A.3.1 PHYSICAL STABILITY

This metric assesses whether the generated trajectories preserve physically stable conformations, which is essential to ensure chemical validity and avoid unrealistic molecular structures. We evaluate stability from two complementary perspectives:

1. **Local Structure Stability.** To assess whether the generated trajectories maintain chemically reasonable local geometries, we calculate the deviations of bond lengths and bond angles with respect to the initial frame of the reference trajectories. Both the Mean Absolute Error (MAE) and Mean Squared Error (MSE) are reported. Lower values indicate that the generated conformations remain close to the idealized covalent structure and are thus more chemically stable.

2. **Steric Clashes.** We further quantify the presence of steric conflicts, which occur when non-bonded atoms are unrealistically close to each other. Specifically, a clash is counted if the interatomic distance (excluding bonded pairs and angle-related atoms) is less than a threshold of $1.5$ Å. We compute clash scores for both intra-ligand and protein–ligand interactions, where the score corresponds to the average number of clashes per generated conformation. Lower clash scores indicate physically more plausible conformations.

### A.3.2 CONFORMATIONAL FLEXIBILITY

In addition to stability, it is important that generated trajectories capture the dynamic flexibility of molecular systems. For the MISATO protein–ligand interaction dataset, we adopt the Root Mean Square Fluctuation (RMSF) to quantify the extent of atomic motion over time after trajectory alignment:

$$\text{RMSF}_i = \sqrt{\frac{1}{T} \sum_{t=1}^{T} \|\mathbf{r}_i(t) - \bar{\mathbf{r}}_i\|^2},$$

where $\mathbf{r}_i(t)$ is the position of atom $i$ at time $t$, and $\bar{\mathbf{r}}_i$ is its time-averaged position.

We evaluate flexibility from two perspectives: 1. **Global Consistency.** We compute the Pearson correlation coefficient between the RMSFs of generated and reference trajectories, where higher correlation indicates better agreement in the fluctuation profiles. 2. **Magnitude Accuracy.** We also report the average RMSF of the generated trajectories. Values closer to the reference average RMSF imply that the model produces realistic levels of conformational motion rather than being overly rigid or excessively flexible.

### A.3.3 UNBINDING PATH DISTANCE

For the DD-13M ligand unbinding dataset, we evaluate whether generated unbinding trajectories follow realistic spatial pathways compared to reference simulations. We compute the Root Mean Square Deviation (RMSD) between generated and reference ligand centroid trajectories with the following procedure:

1. **Trajectory Standardization.** All ligand centroid trajectories are resampled to a uniform length ($L = 100$ frames) using linear interpolation, ensuring comparability between different sequences.

2. **Best-Match Search.** For each generated trajectory, we identify the reference trajectory that yields the minimum RMSD. This accounts for the possibility of multiple plausible unbinding pathways. Let $\mathbf{C}_{\text{gen}} = (\mathbf{c}_1, \ldots, \mathbf{c}_L)$ denote the sequence of ligand centroids in a generated trajectory, and let $\mathcal{T}_{\text{ref}} = \{\mathbf{C}_{\text{ref}}^{(k)}\}_{k=1}^{K}$ be the set of $K$ reference trajectories. The distance metric is calculated as:

$$\text{RMSD}_{\text{best}} = \min_k \sqrt{\frac{1}{L} \sum_{t=1}^{L} \|\mathbf{c}_t - \mathbf{c}_t^{(k)}\|_2^2} \tag{5}$$

To ensure a fair comparison for the static baseline, we construct a pseudo-trajectory by replicating the initial bound pose $L$ times (i.e., $\mathbf{c}_t = \mathbf{c}_1$ for all $t$).

3. **Final Score.** The reported metric is the average of these best-match RMSDs across all generated trajectories. Lower RMSD values indicate that the model generates ligand motions more consistent with physically realistic unbinding paths.

### A.3.4 UNBINDING SUCCESS

This metric evaluates whether the generated ligand trajectories successfully capture the unbinding event. Specifically, we construct the convex hull of the protein heavy atoms in the initial bound state. If at least one predicted ligand centroid position lies outside this convex hull, the trajectory is considered as a successful unbinding case.

We report the **Success@k**, which measures the probability that at least one out of $k$ independently generated trajectories for the same protein–ligand complex achieves successful unbinding. A higher

success rate indicates a better capability of the model to reproduce realistic ligand unbinding processes. Formally, for each complex with $k$ attempts, Success@k is defined as

$$\text{Success@}k = \frac{1}{N} \sum_{n=1}^{N} \mathbb{I}\left[ \max_{1 \leq j \leq k} s_n^{(j)} = 1 \right],$$

where $s_n^{(j)}$ is the binary success indicator (1 if the $j$-th trajectory of complex $n$ achieves unbinding, 0 otherwise), and $N$ is the total number of complexes. We report Success@1, Success@5, and Success@10, which reflect performance under varying generation attempts, respectively.

### A.3.5 UNBINDING CORRECT RATES

To provide a more rigorous evaluation that directly addresses whether our model identifies the correct exit pathway, we perform an analysis on the successful unbinding trajectories. We define the ground-truth exit position, $\mathbf{c}_{\text{exit}}^{\text{GT}}$, as the ligand's centroid coordinates in the final frame of the reference simulation. We introduce two metrics to quantify the spatial accuracy of the generated exit paths:

1. **Endpoint Distance.** This metric measures how closely the generated pathway approaches the known exit location. For a generated trajectory consisting of $L$ frames, denoted as $\mathbf{C}_{\text{gen}} = (\mathbf{c}_1, \ldots, \mathbf{c}_L)$, we calculate the minimum Euclidean distance between any centroid position along the predicted path and the ground-truth exit position:

$$D_{\text{end}} = \min_{t \in \{1, \ldots, L\}} \|\mathbf{c}_t - \mathbf{c}_{\text{exit}}^{\text{GT}}\|_2 \tag{6}$$

   We report the mean of this minimum distance across all test samples. Lower values indicate that the generated trajectory passes closer to the true exit point.

2. **Correct Rate.** To provide an intuitive measure of success, we calculate the percentage of trajectories that successfully navigate within a specific proximity of the correct exit. Given a distance threshold $\tau$ (e.g., $0.5\,\text{Å}$ or $1.0\,\text{Å}$), the correct rate is defined as:

$$\text{Rate}(\tau) = \frac{1}{N} \sum_{i=1}^{N} \mathbb{I}(D_{\text{end}}^{(i)} < \tau) \tag{7}$$

   where $N$ is the total number of samples, $D_{\text{end}}^{(i)}$ is the endpoint distance for the $i$-th sample, and $\mathbb{I}(\cdot)$ is the indicator function.

### A.4 EXPERIMENTAL DETAILS

### A.4.1 SAMPLING SETTINGS

For the first coarse-grained forecasting stage, we set the sampling step $k = 10$ for all three datasets, with different underlying simulation time intervals:

(i) For the MISATO dataset, the all-atom MD simulations were performed with a 80 ps timestep. Therefore, our sampling of every $k = 10$ steps corresponds to a physical time interval of 0.8 ns (10 steps $\times$ 0.08 ns /step).

(ii) For the DD-13M dataset, the simulations utilized a 0.1 ps timestep. Consequently, sampling every k=10 steps resulted in a time interval of 1 ps between saved frames (10 steps $\times$ 0.1 ps/step). For two case studies (6EY8 and 3PCU), we sampled 100 trajectories to compute the unbinding correct rate metric.

(iii) For the Atlas dataset, each trajectory has a total simulation length of 100 ns and is provided as a sequence of 10,000 frames. Consequently, sampling every k=10 steps resulted in a time interval of 1 ns between saved frames (10 steps $\times$ 100 ps/step).

### A.4.2 HYPERPARAMETERS

We use 8 NVIDIA RTX A6000 GPUs for training. For each dataset, we use the Adam optimizer and train BioMD for 20,000 steps with a fixed learning rate of 0.0001 and a batch size of 32. The hyperparameters for `FlowTrajectoryTransformer`, `AtomAttentionEncoder`, `AtomAttentionDecoder`, and `GraphTransformer` is shown in **Table 4,5,6**.

**Table 4.** Key hyperparameters for the `FlowTrajectoryTransformer` model.

| Hyperparameter | Symbol | Value | Description |
|---|---|---|---|
| Number of Transformer Blocks | $N_{\text{block}}$ | 24 | Number of sequential transformer layers. |
| Number of Attention Heads | $N_{\text{head}}$ | 16 | Number of heads in the multi-head attention. |
| Single Representation Dim. | $c_s$ | 384 | Dimension of the per-residue representation. |
| Pair Representation Dim. | $c_z$ | 128 | Dimension of the residue-pair representation. |

**Table 5.** Key hyperparameters for the `AtomAttentionEncoder` and `AtomAttentionDecoder` model.

| Hyperparameter | Symbol | Value | Description |
|---|---|---|---|
| Number of Transformer Blocks | $N_{\text{block}}$ | 3 | Number of sequential transformer layers. |
| Number of Attention Heads | $N_{\text{head}}$ | 4 | Number of heads in the multi-head attention. |
| Single Atom Representation Dim. | $c_{\text{atom}}$ | 128 | Dimension of the per-atom representation. |
| Pair Atom Representation Dim. | $c_{\text{atompair}}$ | 16 | Dimension of the atom-pair representation. |

**Table 6.** Key hyperparameters for the `GraphTransformer` model.

| Hyperparameter | Symbol | Value | Description |
|---|---|---|---|
| Number of Encoder Layers | $L$ | 6 | Number of encoder layers. |
| Encoder Embedding Dimension | $H$ | 384 | Encoder embedding dimension. |
| FFN Embedding Dimension | $F$ | 1536 | Encoder embedding dimension for FFN. |
| Number of Attention Heads | $A$ | 16 | Number of encoder attention heads. |

**Table 7. Ablation study on the MISATO dataset.** Comparison of all methods on physical stability (first six metrics) and conformational flexibility (last four metrics). Mean values on the test samples are reported.

| Method | Bond Geometry[a] | | Angle Geometry[a] | | Steric Clashes | | RMSF Correlation[b] | | RMSF Value[a,c] | |
|---|---|---|---|---|---|---|---|---|---|---|
| | MAE | MSE | MAE | MSE | Intra-Lig | Prot-Lig | Ligand | Protein | Ligand (1.211) | Protein (1.002) |
| Molecular Dynamics | .0377 | .0023 | .0575 | .0053 | 0 | 0 | - | - | - | - |
| BioMD (k=1) | **.0356** | **.0021** | **.0568** | **.0053** | **0** | **0** | .4526 | .6291 | .3717 | .3825 |
| BioMD (k=5) | .0387 | .0024 | .0627 | .0066 | **0** | .0001 | **.4982** | .6023 | .4552 | .4287 |
| BioMD (k=10) | .0495 | .0155 | .0709 | .0097 | .0019 | .0023 | .4789 | **.6854** | **.7023** | **.6242** |
| BioMD (k=20) | .0421 | .0030 | .0698 | .0087 | .0002 | .0008 | .4267 | .5309 | .5403 | .4959 |

[a] Bond geometry (bond length) and RMSF values are in angstroms (Å). Angle geometry (bond angle) is in radians.
[b] RMSF Correlation is reported using the Pearson correlation coefficient.
[c] RMSF values for reference trajectories are given in parentheses. Values closer to those of the reference indicate better results.

**Table 8. Ablation study on the DD-13M dataset.** Comparison of methods on physical stability (first six metrics), ligand unbinding path reconstruction metric (Unbinding Path RMSD), and ligand unbinding success rates. Mean values on the test samples are reported.

| Method | Bond Geometry[a] | | Angle Geometry[a] | | Steric Clashes | | Unbinding Path[a] | Unbinding Success | | |
|---|---|---|---|---|---|---|---|---|---|---|
| | MAE | MSE | MAE | MSE | Intra-Lig | Prot-Lig | RMSD | @1 | @5 | @10 |
| Static | - | - | - | - | .2778 | 0 | .6504 | 0 | 0 | 0 |
| Metadynamics[b] | .0246 | .0012 | .0452 | .0030 | .2777 | 0 | .4217 | - | - | - |
| BioMD (AR-1, k=10) | .1880 | .0612 | .2011 | .0761 | 1.727 | 4.079 | 1.131 | .2285 | .4728 | .5822 |
| BioMD (AR-5, k=10) | .0728 | .0111 | .0802 | .0132 | **.2943** | .0009 | **.5645** | .5676 | .7419 | .7941 |
| BioMD (AR-10, k=10) | .1067 | .0221 | .1478 | .0450 | .7098 | .5356 | .7465 | .4008 | .5719 | .6239 |
| BioMD (w.o. AR, k=10) | **.0308** | **.0018** | **.0606** | **.0077** | .2944 | **.0004** | .6845 | .0029 | .0147 | .0294 |
| BioMD (AR-5, k=10, L*2)[c] | .1457 | .0382 | .1770 | .0595 | 1.407 | 2.029 | .7766 | **.5735** | **.7512** | **.8010** |
| BioMD (AR-5, k=1)[d] | .0682 | .0179 | .1231 | .0321 | .3686 | 7.380 | .7313 | .3428 | .5128 | .6293 |
| BioMD (AR-5, k=5) | .0575 | .0164 | .1062 | .0246 | .4700 | 1.346 | .5703 | .5214 | .6582 | .7023 |
| BioMD (AR-5, k=10) | .0728 | .0111 | .0802 | .0132 | **.2943** | **.0009** | **.5645** | **.5676** | **.7419** | .7941 |
| BioMD (AR-5, k=20) | **.0497** | **.0053** | .0914 | .0184 | .3736 | .0967 | .5909 | .4714 | .7065 | **.8043** |

[a] Bond geometry (bond length) and unbinding path RMSD values are in angstroms (Å), and angle geometry (bond angle) is in radians.
[b] The metadynamics trajectory serves as the lower-bound. The metrics are calculated among trajectories of multiple repeating simulations.
[c] "L*2" means that the generated trajectory length is doubled (i.e., 1000 frames).
[d] $k = 1$ represents the "forecasting-only" version of BioMD.

A.5    ABLATION STUDY

We conducted ablation studies using BioMD-rel, generating 8 ns trajectories for MISATO and 500 frames for DD-13M. During forecasting, we employed a moving window of the 50 most recent frames as historical context to mitigate GPU memory constraints. The quantitative results, presented in **Tables 7 and 8**, demonstrate that the model's performance is highly sensitive to both the auto-regressive block size (AR-$x$), which is the number of concurrently denoised frames during sampling, and the hierarchical step size ($k$), which is the frame interval in the coarse-grained forecasting stage.

We first observe that the auto-regressive block size is a critical determinant of physical stability. A minimal block size (e.g., AR-1) necessitates a high number of sequential inference steps, leading to severe error accumulation. This accumulation manifests as catastrophic physical violations, with protein-ligand clashes increasing by over 4000 times compared to the baseline. Conversely, an excessively large block size (e.g., w.o. AR) appears to increase the task complexity, confusing the model and degrading predictive accuracy. Consequently, AR-5 emerges as the optimal trade-off, providing sufficient context for stable predictions without introducing unnecessary computational difficulty.

Furthermore, a larger step size ($k$) proves essential for mitigating error propagation. Our experiments show that a non-hierarchical approach ($k = 1$) suffers from extreme instability due to the vast number of required inference steps, resulting in protein-ligand clashes exceeding 7000 times the baseline levels. While the model is generally less sensitive to $k$ than to the AR setting, the results indicate that $k = 10$ yields superior efficacy for unbinding tasks. Therefore, the specific combination of AR-5 and $k = 10$ was selected as the final configuration, as it effectively balances unbinding metrics with the maintenance of physical realism in complex biomolecular dynamics.

A.6    ADDITIONAL EXPERIMENTAL RESULTS

A.6.1    BOND AND ANGLE ERROR

To evaluate the geometric accuracy and temporal stability of our generative model, we analyzed the distributions of bond length and angle errors, as well as the error accumulation problem. All experiments are conducted with BioMD-rel. The results in **Figure 7** show that on the MISATO test set, 97.9% of bond length errors are below 0.2 Å, and 95.3% of bond angle errors are below 0.2 radians ( 11.5°). The performance on the DD-13M test set also demonstrates 99.7% of bond errors and 98.2% of angle errors falling below these thresholds.

Furthermore, we examined the trend of errors over time. On the MISATO dataset, due to the short generated trajectories (within 100 frames), the errors across different time spans are similar. In contrast, on the DD-13M dataset involving long trajectory generation, as the number of generated frames increases (up to 500 frames), a significant increase in error is observed, indicating the occurrence of the error accumulation problem.

A.6.2    RELAXATION CONSISTENCY ANALYSIS

To assess the energetic quality of the generated conformations, we conducted a relaxation consistency analysis on them. For each trajectory generated by BioMD-rel on the MISATO test set, we took each frame and performed energy minimization using the Amber99SB force field. We then calculated the all-atom RMSD between the pre- and post-relaxation structures. A low RMSD indicates that our generated conformations are near a local energy minimum, confirming their physical plausibility. Our analysis reveals a mean all-atom RMSD of 0.72 Å  (**Figure 8**) after relaxation across all generated frames (0.69 Å  for MD conformations, **Figure 8**), demonstrating the ability of BioMD to generate energetically favorable structures and providing strong evidence for the physical plausibility of our generated trajectories beyond purely geometric metrics.

A.6.3    POCKET-LIGAND INTERACTION ANALYSIS

To evaluate the accuracy of the predicted binding interactions, we perform an analysis on the MISATO test set (1,031 complexes), focusing on the conformation of key binding residues and their interactions with the ligand. We identified residues located within 5 Å  of the ligand and compared

their conformational properties against ground truth MD simulations. All experiments are conducted with BioMD-rel. As shown in **Figure 9**, the chi1 dihedral angle distributions predicted by BioMD (red dashed line) closely align with those observed in the MD trajectories (blue solid line). Furthermore, we visualized the interaction patterns by calculating the contact probability between protein residues and ligand atoms using a 5 Å cutoff (**Figure 10**). The resulting contact probability map from BioMD exhibits a high degree of consistency with the MD reference, confirming that the model accurately captures both the side-chain geometries and the dynamic contact profiles within the binding pocket.

### A.6.4 TICA ANALYSIS

For each protein in the test set, we projected the generated conformations onto a 2D space using Time-lagged Independent Component Analysis (TICA). We train the BioMD-rel model on the Atlas dataset, which contains 100 ns MD trajectories for 1,390 protein chains. The first two independent components (tIC1 and tIC2), which capture the slowest and most significant motions of the system, were used to define a 2D conformational space. We first fit TICA on a 100 ns MD trajectory and then projected the conformations generated by BioMD onto this 2D space. Specifically, we used BioMD to sample three 100 ns trajectories and projected all generated conformations onto the TICA space. For comparison, we also projected conformations from other MD simulation replicas (100 ns, 10 ns, and 1 ns) onto the same space. As shown in **Figure 13**, BioMD explores the sample space more sufficiently than 1 ns and 10 ns MD simulations, capturing a broader range of conformational diversity. Notably, the average sampling time for one trajectory is only 56 seconds, making BioMD much faster than MD simulations.

**Table 9. Comparison of methods on endpoint distance and unbinding correct rates.** Mean values on the test samples are reported.

| Method | Endpoint Distance | Correct Rate (< X Å) | | | |
|---|---|---|---|---|---|
| | RMSD (Å) | < 0.5 Å | < 1 Å | < 2 Å | < 4 Å |
| Static | 3.5852 | 0.1765 | 0.3235 | 0.4118 | 0.7353 |
| Metadynamics[b] | 0.9739 | 0.5136 | 0.7532 | 0.9022 | 0.9641 |
| BioMD-rel | 2.3433 | 0.4265 | 0.5265 | 0.6971 | 0.8000 |
| BioMD-abs | 2.3897 | 0.3765 | 0.5324 | 0.6882 | 0.8059 |
| BioMD-rel (AR-5) | 1.8001 | 0.3471 | 0.5882 | 0.7529 | **0.8706** |
| BioMD-abs (AR-5) | **1.7253** | **0.4912** | **0.6824** | **0.7853** | 0.8588 |

[b] The metadynamics trajectory serves as the lower-bound. The metrics are calculated among trajectories of multiple repeating simulations.

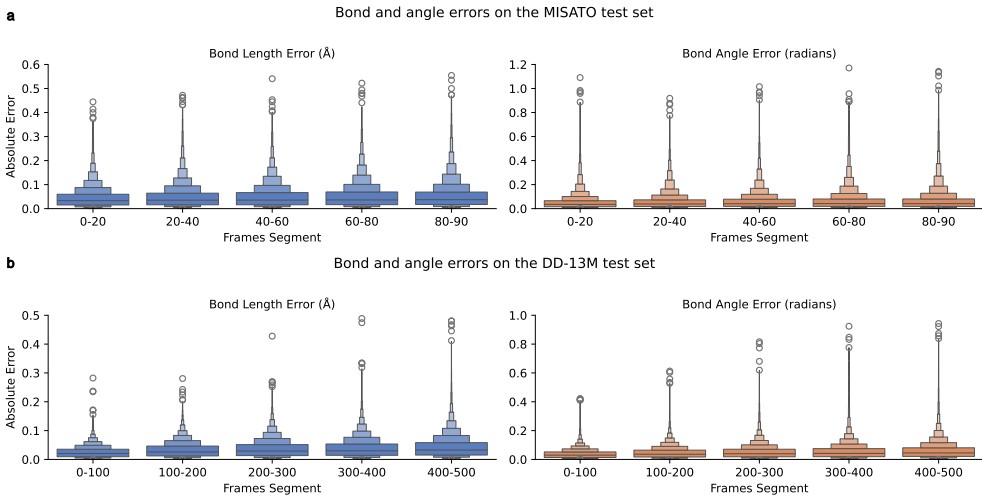

**Figure 7. Bond and angle error distribution**. **a.** Boxen plot showing error distribution on the MISATO test set. **b.** Boxen plot showing error distribution on the DD-13M test set.

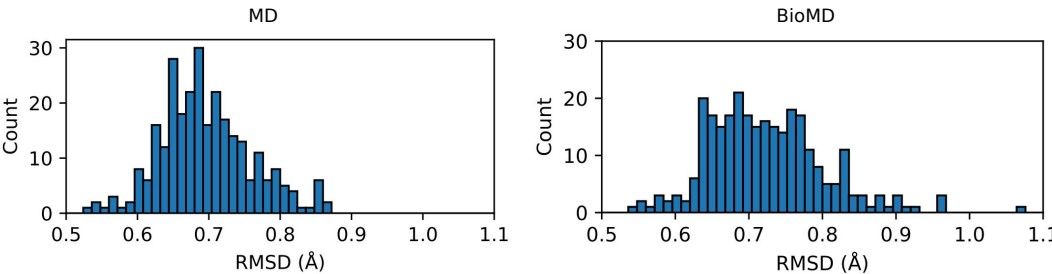

**Figure 8. All-atom RMSD after relaxation**. All-atom RMSD when using the Amber99SB force field for relaxation on MD conformations (mean=0.69 Å ) and BioMD-generated conformations (mean=0.72 Å ).

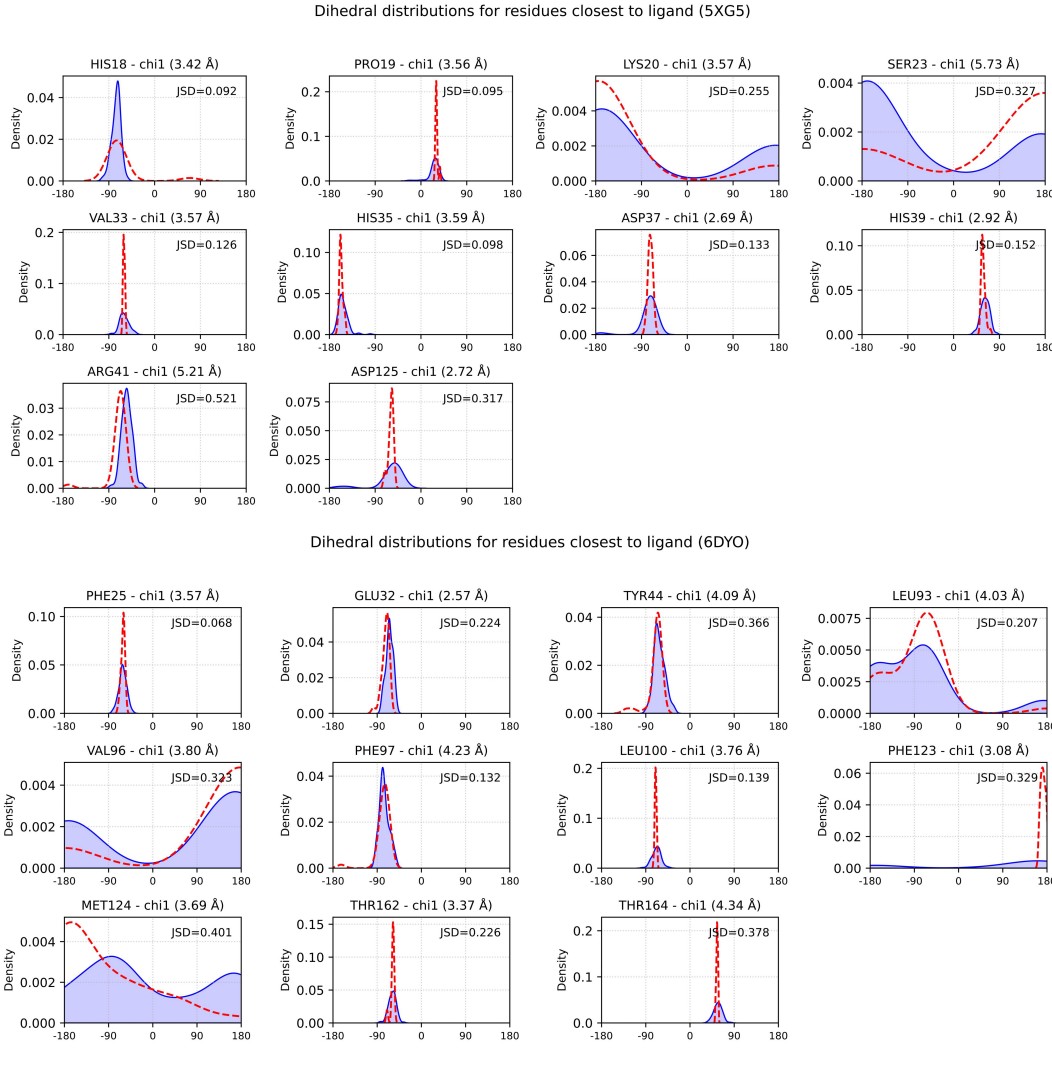

**Figure 9. Dihedral distributions for residues closest to the ligand.** The dashed red line represents BioMD, while the solid blue line represents the ground truth MD simulations. The value in parentheses indicates the minimum distance between the considered residue atoms and all ligand atoms. Test cases (6DYO and 5XG5) are from the MISATO test set.

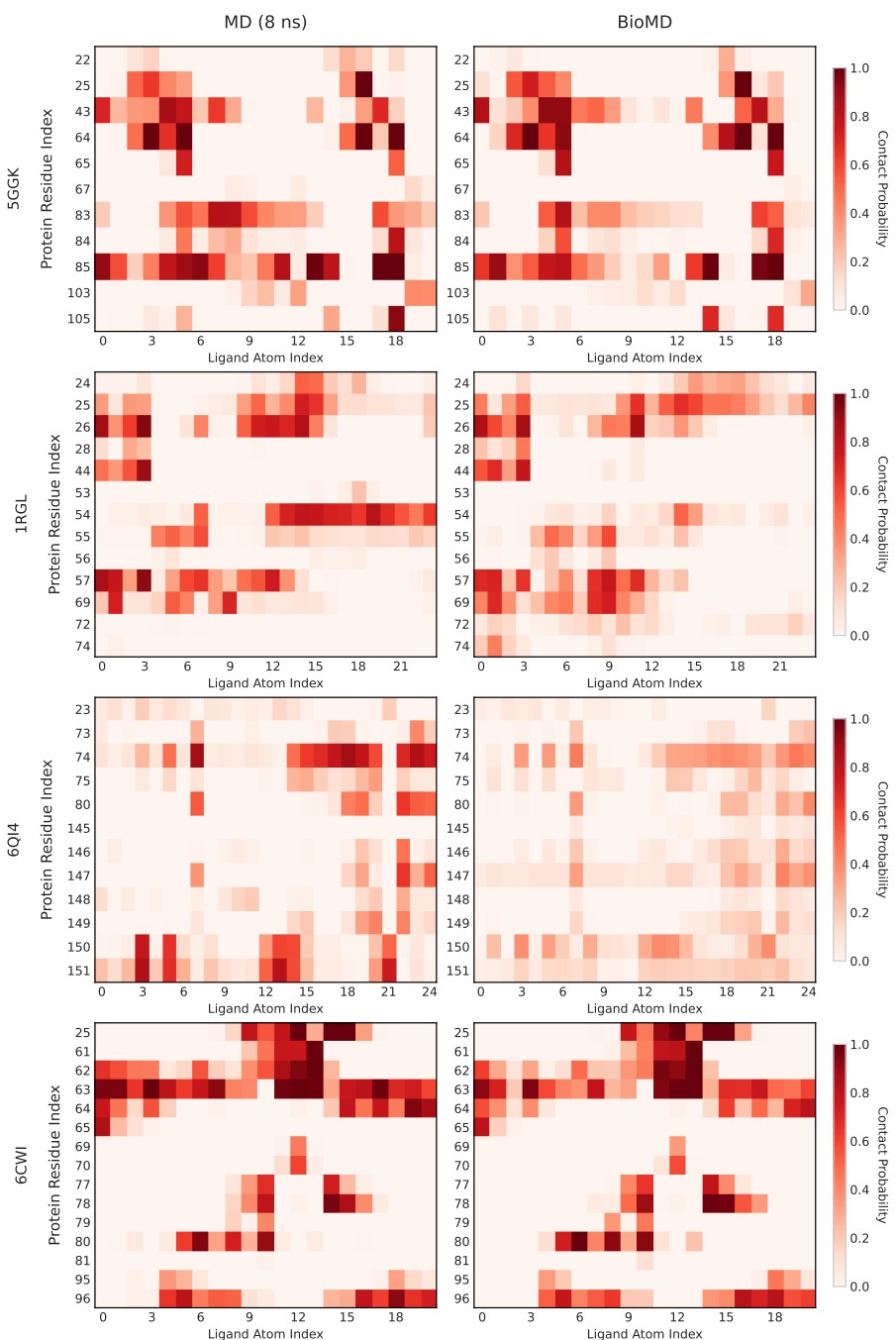

**Figure 10. Protein-ligand contact probability maps.** The heatmaps compare the protein-ligand contact probability maps between MD reference (left) and BioMD (right), calculated using a 5 Å distance cutoff. Test cases (5GGK, 1RGL, 6QI4, and 6CWI) are from the MISATO test set.

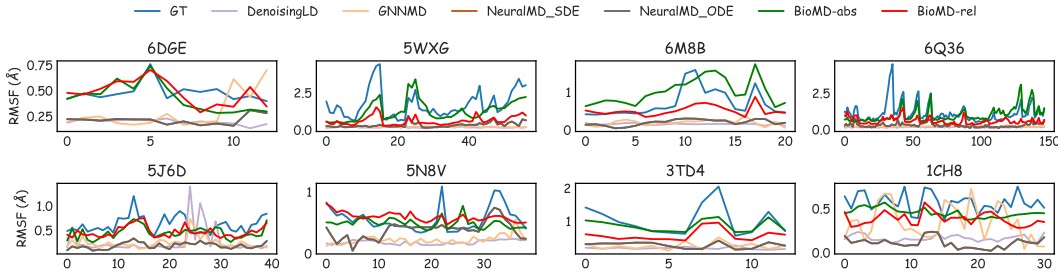

**Figure 11. Ligand RMSF on the MISATO dataset**. Line plot showing Ligand RMSF for eight different protein-ligand systems from the MISATO test set.

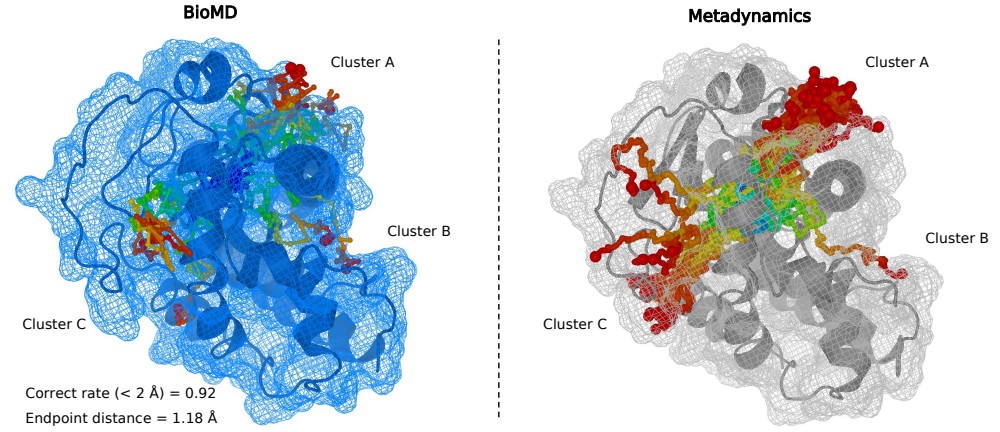

**Figure 12. OOD case study**. The test case (3PCU) comes from the DD-13M test set, with an unbinding path RMSD of 0.77 Å (Metadynamics=0.65 Å). Its largest sequence similarity with the training set is 0.26.

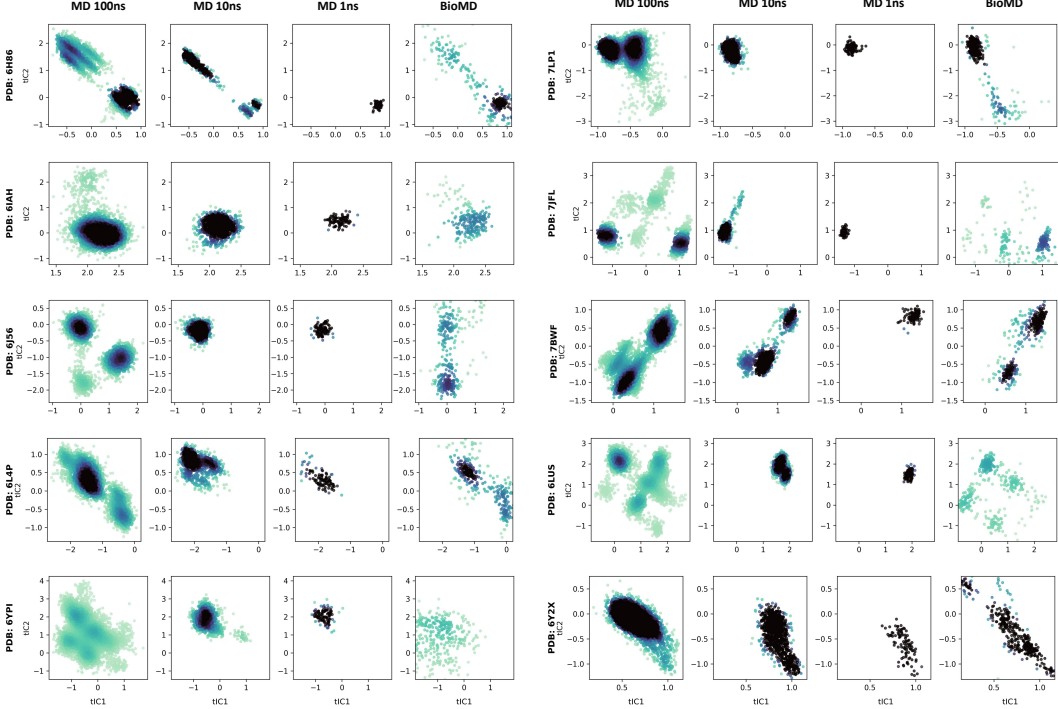

**Figure 13. TICA plot on the Atlas test set**. Time-lagged Independent Component Analysis (TICA) plot showing 2D conformational space for MD (100 ns, 10 ns, and 1 ns) and BioMD. we use dihedral angles of the backbone ($\phi, \psi$) as features for TICA. Test cases are from the Atlas test set.

---

**Algorithm 1:** Main Inference Loop

---

**Input:** $\{\mathbf{f}^*\}$, $\{\vec{\mathbf{x}}_{0,l}\}$, $N_{\text{cycle}} = 4$, $c_s = 384$, $c_z = 128$

1  $\{\mathbf{s}_i^{\text{inputs}}\} \leftarrow$ InputFeatureEmbedder($\{\mathbf{f}^*\}$);

2  $\mathbf{s}_i^{\text{init}} \leftarrow$ LinearNoBias($\mathbf{s}_i^{\text{inputs}}$);

3  $\mathbf{z}_{ij}^{\text{init}} \leftarrow$ LinearNoBias($\mathbf{s}_i^{\text{inputs}}$) + LinearNoBias($\mathbf{s}_j^{\text{inputs}}$);

4  $\{\mathbf{z}_{ij}\}, \{\mathbf{s}_i\} \leftarrow 0, 0$;

5  **foreach** $c \in \{1, \dots, N_{cycle}\}$ **do**

6  $\quad$ $\mathbf{z}_{ij} \leftarrow \mathbf{z}_{ij}^{\text{init}} +$ LinearNoBias(LayerNorm($\mathbf{z}_{ij}$));

7  $\quad$ $\{\mathbf{z}_{ij}\}, \{\mathbf{s}_i\} \leftarrow$ GraphTransformer($\{\vec{\mathbf{x}}_{0,l}\}, \{\mathbf{s}_i\}, \{\mathbf{z}_{ij}\}, \{\mathbf{s}_i^{\text{inputs}}\}$);

8  $\quad$ $\mathbf{s}_i \leftarrow \mathbf{s}_i^{\text{init}} +$ LinearNoBias(LayerNorm($\mathbf{s}_i$));

9  traj_list=$[\{\vec{\mathbf{x}}_{0,l}\}]$ ;

10  **foreach** $t \in \{1, \dots, T\}$ **do**

11  $\quad$ $\{\vec{\mathbf{x}}_{t,l}^{\text{pred}}\} \leftarrow$ SampleFlow($\{\vec{\mathbf{x}}_{his}\}, \{\mathbf{f}^*\}, \{\mathbf{s}_i^{\text{inputs}}\}, \{\mathbf{s}_i\}, \{\mathbf{z}_{ij}\}$);

12  $\quad$ traj_list.add($\vec{\mathbf{x}}_t^{\text{pred}}$);

13  $\quad$ $\{\vec{\mathbf{x}}_{his}\}$ = traj_list;

14  **return** *traj_list*

---

**Algorithm 2:** TrainFlow

---

**Input:** $\{\vec{\mathbf{x}}_l\}, \{\vec{\mathbf{x}}_{his}\}, \{\mathbf{f}^*\}, \{\mathbf{s}_i^{\text{inputs}}\}, \{\mathbf{s}_i^{\text{trunk}}\}, \{\mathbf{z}_{ij}^{\text{trunk}}\}$

1  # Indepentent noise levels ;

2  $\tau \sim (\mathcal{U}(0,1), \mathcal{U}(0,1), \cdots, \mathcal{U}(0,1))$;

3  $\{\vec{\mathbf{x}}_l^0\} \sim \mathcal{N}(\vec{0}, \mathbf{I}_3)$;

4  $\{\vec{\mathbf{x}}_l\} \leftarrow$ CentreRandomAugmentation($\{\vec{\mathbf{x}}_l\}$);

5  $\{\vec{\mathbf{x}}_l^\tau\} = \tau\{\vec{\mathbf{x}}_l\} + (1-\tau)\{\vec{\mathbf{x}}_l^0\}$;

6  $\{\vec{\mathbf{u}}_l^\tau\} \leftarrow$ FlowModule($\{\vec{\mathbf{x}}_l^\tau\}, \{\vec{\mathbf{x}}_{his}\}, \tau, \{\mathbf{f}^*\}, \{\mathbf{s}_i^{\text{inputs}}\}, \{\mathbf{s}_i^{\text{trunk}}\}, \{\mathbf{z}_{ij}^{\text{trunk}}\}$);

7  $\mathcal{L}_{flow} =$ MSE($\{\vec{\mathbf{u}}_l^\tau\}, \{\frac{\vec{\mathbf{x}}_l - \vec{\mathbf{x}}_l^\tau}{1-\tau}\}$);

8  **return** $\mathcal{L}_{flow}$

---

**Algorithm 3:** SampleFlow

---

**Input:** $\{\vec{\mathbf{x}}_{his}\}, \{\mathbf{f}^*\}, \{\mathbf{s}_i^{\text{inputs}}\}, \{\mathbf{s}_i^{\text{trunk}}\}, \{\mathbf{z}_{ij}^{\text{trunk}}\}$

1  $\vec{\mathbf{x}}_l^0 \sim \mathcal{N}(\vec{0}, \mathbf{I}_3)$;

2  **foreach** $\tau$ *in* $\{0, 0.1, 0.2, ..., 0.9\}$ **do**

3  $\quad$ $\{\vec{\mathbf{u}}_l^\tau\} \leftarrow$ FlowModule($\{\vec{\mathbf{x}}_l^\tau\}, \{\vec{\mathbf{x}}_{his}\}, \tau, \{\mathbf{f}^*\}, \{\mathbf{s}_i^{\text{inputs}}\}, \{\mathbf{s}_i^{\text{trunk}}\}, \{\mathbf{z}_{ij}^{\text{trunk}}\}$);

4  $\quad$ $\vec{\mathbf{x}}_l^{\tau+1} \leftarrow \vec{\mathbf{x}}_l^\tau + dt \cdot \vec{u}_l^\tau$;

5  **return** $\{\vec{\mathbf{x}}_l^1\}$

---

**Usage Note:** The returned bias is added to the attention logits inside the main transformer block before the softmax activation: logits $\leftarrow (Q \cdot K^T)/\sqrt{d_k} +$ bias.

---

**Algorithm 4:** FlowModule

---

**Input:** $\{\vec{\mathbf{x}}_l^{\text{noisy}}\}, \{\vec{\mathbf{x}}_{his}\}, t, \{\mathbf{f}^*\}, \{\mathbf{s}_i^{\text{inputs}}\}, \{\mathbf{s}_i^{\text{trunk}}\}, \{\mathbf{z}_{ij}^{\text{trunk}}\}$
$\sigma_{\text{data}} = 16, c_{\text{atom}} = 128, c_{\text{atompair}} = 16, c_{\text{token}} = 768$

1   $\{\mathbf{s}_i\}, \{\mathbf{z}_{ij}\} \leftarrow$ FlowConditioning$(t, \{\mathbf{f}^*\}, \{\mathbf{s}_i^{\text{inputs}}\}, \{\mathbf{s}_i^{\text{trunk}}\}, \{\mathbf{z}_{ij}^{\text{trunk}}\}, \sigma_{\text{data}})$;

2   # Sequence-local Atom Attention with history info and aggregation to coarse-grained tokens ;

3   $\{a_i\}, \{q_k^{\text{skip}}\}, \{p_k^{\text{skip}}\}, \{t_k^{\text{skip}}\} \leftarrow$
     AtomAttentionEncoder$(\{\vec{\mathbf{x}}_{his}\}, \{\mathbf{f}^*\}, \{\vec{\mathbf{x}}_l^{\text{noisy}}\}, \{\mathbf{s}_i\}, \{\mathbf{z}_{ij}\}, c_{\text{atom}}, c_{\text{atompair}}, c_{\text{token}})$;

4   # Full self-attention on token level.;

5   $a_i \leftarrow$ LinearNoBias(LayerNorm($a_i$));

6   $\{a_k\} \leftarrow$ FlowTrajectoryTransformer$(\{a_i\}, \{\mathbf{s}_i\}, \{\mathbf{z}_{ij}\}, \beta_{ij} = 0, N_{\text{block}} = 24, N_{\text{head}} = 16)$;

7   $a_i \leftarrow$ LayerNorm($a_i$);

8   # Broadcast token activations to atoms and run Atom Attention.;

9   $\{\vec{\mathbf{u}}_l\} \leftarrow$ AtomAttentionDecoder$(\{a_i\}, \{q_k^{\text{skip}}, p_k^{\text{skip}}, t_k^{\text{skip}}\})$;

10   **return** $\{\vec{\mathbf{u}}_l\}$

---

**Algorithm 5:** FlowConditioning

---

**Input:** $\hat{t}, \{\mathbf{f}^*\}, \{s_i^{\text{inputs}}\}, \{s_i^{\text{trunk}}\}, \{z_{ij}^{\text{trunk}}\}, \sigma_{\text{data}}, c_z = 128, c_s = 384$

1   # Pair conditioning;

2   $z_{ij} \leftarrow$ LinearNoBias(LayerNorm($z_{ij}$));

3   **foreach** $b \in \{1, 2\}$ **do**

4     $z_{ij}$ += Transition$(z_{ij}, \; n = 2)$;

5   # Single conditioning;

6   $s_i \leftarrow$ concat($[\; s_i^{\text{trunk}}, \; s_i^{\text{inputs}} \;]$);

7   $s_i \leftarrow$ LinearNoBias(LayerNorm($s_i$));

8   **foreach** $b \in \{1, 2\}$ **do**

9     $s_i$ += Transition$(s_i, \; n = 2)$;

10   **return** $\{s_i\}, \{z_{ij}\}$

---

**Algorithm 6:** FlowTrajectoryTransformer

---

**Input:** $\{a_i\}, \{s_i\}, \{z_{ij}\}, \{\beta_{ij}\}, N_{\text{block}} = 24, N_{\text{head}} = 16$

1   **for** $n \in [1, \ldots, N_{\text{block}}]$ **do**

2     $\{b_i\} \leftarrow$ AttentionPairBias$(\{a_i\}, \{s_i\}, \{z_{ij}\}, \{\beta_{ij}\}, N_{\text{head}})$;

3     $\{b_i\} \leftarrow$ TemporalAttention$(\{a_i + b_i\})$;

4     $a_i \leftarrow b_i +$ ConditionedTransitionBlock$(a_i, s_i)$;

5   **return** $\{a_i\}$

---

---

**Algorithm 7:** `TemporalAttention` Module

---

**Input:** Single representation $c_s$ of shape $(T, N, c_s)$, where $T$ is time, $N$ is residues.
**Output:** Updated single representation output of shape $(T, N, c_s)$.

1 # Permute dimensions to make time the sequence axis for attention
    $c_s' \leftarrow \text{Permute}(c_s, \text{dims} = (1, 0, 2))$;
2 # Project to Query, Key, Value for each residue independently;
3 $Q \leftarrow \text{Linear}_Q(c_s')$;
4 $K \leftarrow \text{Linear}_K(c_s')$;
5 $V \leftarrow \text{Linear}_V(c_s')$;
6 # Calculate scaled dot-product attention scores across time;
7 $d_k \leftarrow$ dimension of $K$;
8 $\text{logits} \leftarrow (Q \cdot K^T)/\sqrt{d_k}$;
9 $\text{weights} \leftarrow \text{Softmax}(\text{logits}, \text{dim} = -1)$;
10 # Apply attention weights to values;
11 $\text{output}' \leftarrow \text{weights} \cdot V$;
12 # Permute back to the original dimension order;
13 $\text{output} \leftarrow \text{Permute}(\text{output}', \text{dims} = (1, 0, 2))$;
14 **return** *output*;

---

**Algorithm 8:** `AttentionPairBias` Module

---

**Input:** Pair representation $c_z$ of shape $(N, N, c_z)$, where $N$ is residues.
**Output:** Attention bias bias of shape $(N_{\text{head}}, N, N)$.

1 # Project $c_z$ to match the number of attention heads;
2 $\text{bias}' \leftarrow \text{Linear}_{\text{bias}}(c_z)$;
3 # Permute dimensions to align with attention logits shape;
4 $\text{bias} \leftarrow \text{Permute}(\text{bias}', \text{dims} = (2, 0, 1))$;
5 **return** *bias*;

---

