# OpenReview forum: "BioMD: All-atom Generative Model for Biomolecular Dynamics Simulation"
_ICLR.cc/2026/Conference — ICLR 2026 Poster_

### Official Review · Reviewer_gM8W · 2025-10-26

**Soundness:** 3
**Presentation:** 4
**Contribution:** 2
**Rating:** 4
**Confidence:** 5

**Summary:**

This paper develops a method BioMD, which generates time evolution of protein and ligand structures using a hierarchical architecture and flow matching. BioMD combined coarse grained time interval for structure ensemble generation conditioned on an initial frame, and fine grained interpolation conditioned on the starting and ending frames. It has demonstrated performance on MD datasets MISATO (unbiased protein-ligand simulations) and DD-13M (enhanced sampling MD simulations for ligand unbinding events), and shown advantage in the physicality, flexibility and unbinding success.

**Strengths:**

- The model architecture is novel, proposing a hierarchical framework in combination with flow matching
- The authors have tested multiple different ways of generation, including predicting relative conformational change v.s. absolute conformations, and predicting all frames v.s. auto-regressively, providing a thorough examination
- The BioMD method has shown advantages in the MISATO and DD-13M benchmarks, in the physicality, flexibility and unbinding success.

**Weaknesses:**

- In the introduction, the paper states: "The core bottleneck lies in the intensive calculation of non-bonded forces, particularly van der Waals and electrostatic interactions, which scale quadratically with the number of atoms." This is misleading, because with methods like Particle Mesh Ewald (PME) the scaling is approximately O(N log N), not O(N²).
- The model is trained with a fixed time interval between frames (k=10 steps in coarse-graining). Can the model generalize to different time intervals, or is it restricted to this specific temporal resolution? If the latter, this should be explicitly stated as a limitation.
- The DD-13M dataset is generated using enhanced sampling simulation, instead of unbiased MD. Therefore, BioMD is learning to reproduce biased trajectories, not true thermodynamic/kinetic unbinding behavior. The generated trajectories may not reflect realistic dissociation kinetics, and the model may be learning to mimic the "push" behavior by the biased potential in enhanced sampling. Currently, the paper does not clearly distinguish this subtlety.
- The success criterion (Section A.3.4) only requires that "at least one predicted ligand centroid position lies outside this convex hull." The success rate is thus measuring "can find a way out" rather than "finds the correct way out."
- The reported MAE and MSE metrics average over all frames and trajectories. This can mask catastrophic failures in individual frames. The authors should consider providing more comprehensive failure analysis, including per-frame error distributions, percentage of frames with physical violations etc.
- The paper proposes the hierarchical architecture, but lacks the justification and ablation on the necessity of the hierarchy. Specifically, why the fine-grained generation is needed and how it will influence the performance should be demonstrated and discussed.

**Questions:**

- Section A.3.1 states: "we calculate the deviations of bond lengths and bond angles with respect to the initial frame of the reference trajectories." However, in Section 5, the "Static" baseline is defined as "the initial conformation of the system is held constant throughout the entire trajectory." If errors are computed relative to the initial frame, the Static model should have exactly zero error, yet Table 1 and Table 2 report non-zero errors for Static. The authors need to clarify: (1) what exactly is the reference for computing these errors, and (2) whether errors are computed for ligand only, protein only, or both.
- The phrase "sampling every k = 10 steps" (Section 4.2.1) is imprecise without specifying the timestep used in the underlying simulations. The actual time interval (in ps or fs) should be stated explicitly.

---

> ### Author Response · Authors · 2025-11-26
>
> We sincerely thank the reviewer for the rigorous and insightful feedback. In the revised manuscript, we have corrected the inaccuracies regarding MD complexity and clarified the experimental settings. Additionally, we have included new analyses on error distributions and unbinding pathways to further validate our model's performance and design choices. Our point-by-point responses are listed below.
>
> > **Weakness 1:** In the introduction, the paper states: "The core bottleneck lies in the intensive calculation of non-bonded forces... which scale quadratically with the number of atoms." This is misleading, because with methods like Particle Mesh Ewald (PME) the scaling is approximately O(N log N), not O(N²).
>
> **A1:** Thank you for your careful review!!! We have revised the text to "Although algorithms like Particle Mesh Ewald (PME) have improved scaling to approximately O(N log N), these calculations remain the most computationally demanding component".
>
>
>
> > **Weakness 2:** The model is trained with a fixed time interval between frames (k=10 steps in coarse-graining). Can the model generalize to different time intervals, or is it restricted to this specific temporal resolution? If the latter, this should be explicitly stated as a limitation.
>
> **A2:** Thank you for the question! Our model is not restricted to a temporal resolution of `k=10`. This value was chosen as an optimal hyperparameter through systematic ablation studies, not because of a model limitation. We have clarified this in the ablation study (**Appendix A.5**) of our revised manuscript. Please go to **Common Response 2** for a detailed response.
>
>
>
> > **Weakness 3:** The DD-13M dataset is generated using enhanced sampling simulation... BioMD is learning to reproduce biased trajectories, not true thermodynamic/kinetic unbinding behavior... Currently, the paper does not clearly distinguish this subtlety.
>
> **A3:** This is a crucial point of clarification, and we thank the reviewer for raising it. Our goal with the DD-13M experiment was not to claim that we are learning the true, unbiased kinetics of unbinding. Rather, it was to test whether our model has the capacity to learn and generate the very large-scale, complex conformational changes that are characteristic of a complete unbinding event. Enhanced sampling trajectories provide a data-rich source for such events. We revise the manuscript (**Section 5.2**) to make this distinction crystal clear, emphasizing that the model learns to reproduce the pathways shown in the metadynamics simulations, which do not necessarily represent true thermodynamic or kinetic behavior.
>
>
>
> > **Weakness 4:** The success criterion (Section A.3.4) only requires that "at least one predicted ligand centroid position lies outside this convex hull." The success rate is thus measuring "can find a way out" rather than "finds the correct way out."
>
> **A4:** We sincerely thank the reviewer for this important suggestion! To provide a more rigorous evaluation that directly addresses whether our model "finds the correct way out," we have performed a new analysis on predicted unbinding trajectories. Please go to **Common Response 5** for a detailed response.
>
> > **Weakness 5:** The reported MAE and MSE metrics average over all frames and trajectories. This can mask catastrophic failures in individual frames. The authors should consider providing more comprehensive failure analysis...
>
> **A5:** Thank you for this insightful suggestion! We agree that average metrics can obscure important details about error distribution. To provide a more comprehensive analysis, we add a boxen plot (**Figure 7**) in the revised version, which visualizes the distribution of ligand bond length and bond angle errors across different segments of the trajectories on both the MISATO and DD-13M test sets.
>
> The results show that on the MISATO test set, 97.9% of bond length errors are below 0.2 Å, and 95.3% of bond angle errors are below 0.2 radians (~11.5°). The performance on the DD-13M test set also demonstrates 99.7% of bond errors and 98.2% of angle errors falling below these thresholds.
>
> Furthermore, we examined the trend of errors over time. On the MISATO dataset, due to the short generated trajectories (within 100 frames), the errors across different time spans are similar. In contrast, on the DD-13M dataset involving long trajectory generation, as the number of generated frames increases (up to 500 frames), a significant increase in error is observed, indicating the occurrence of the error accumulation problem as we discussed in **Section 5.4** of our paper.

---

> ### Author Response · Authors · 2025-11-26
> **Responses to questions and weakness 6**
>
> > **Weakness 6:** The paper proposes the hierarchical architecture, but lacks the justification and ablation on the necessity of the hierarchy. Specifically, why the fine-grained generation is needed and how it will influence the performance should be demonstrated and discussed.
>
> **A6:**  Thank you for your feedback. This is a key point, as the hierarchical design is a core contribution. We conducted an ablation study in **Appendix A.5** to demonstrate the necessity of our hierarchical architecture.
>
> In our experiments (**Tables 7, 8**), the `k=1` version represents a "Forecasting-Only" model, which omits the fine-grained interpolation and refinement. Through comparison, we can observe:
>
> 1.  **Ensuring Physical Plausibility:** As shown in results on DD-13M (**Table 8**), the Forecasting-Only model (`k=1`) produced a significant number of steric clashes (protein-ligand clashes: `7.380`). In contrast, with the hierarchical approach (e.g., `k=10`), this issue was almost completely eliminated (`0.0009`). This demonstrates that the fine-grained steps are crucial for generating physically plausible conformations.
>
> 2.  **Improving Trajectory Accuracy and Task Performance:** As shown in results on DD-13M (**Table 8**), the full model (`k=10`) achieved a lower Path-RMSD (`.5645` vs. `.7313`) and a higher dissociation success rate (@10: `.7941` vs. `.6293`) compared to the Forecasting-Only model (`k=1`). This demonstrates that the fine-grained refinement significantly improves the overall quality of the predicted trajectories.
>
> > **Q1:** Section A.3.1 states: "we calculate the deviations of bond lengths and bond angles with respect to the initial frame of the reference trajectories." However, in Section 5, the "Static" baseline is defined as "the initial conformation of the system is held constant..." If errors are computed relative to the initial frame, the Static model should have exactly zero error... Clarify: (1) what exactly is the reference... and (2) whether errors are computed for ligand only, protein only, or both.
>
> **A1**: We thank the reviewer for pointing out this issue. You are correct about the inconsistency, which arose because we used two different evaluation contexts without sufficient clarification.
>
> 1.  **What is the reference?** In both Table 1 and 2, the reference is the **initial frame**.
> *   For **trajectory prediction (Table 1)**, the result is computed between the reference and each frame of the MD, so its error is non-zero and serves as a lower bound of the error. We have corrected it as *Molecular Dynamics* in the revised version.
>
> *   For **single conformation generation (Table 2)**, in the result of the "Static" baseline, both the reference and the sample is the **initial frame**. Thus, the "Static" model's error is indeed zero.
>
> 2.  **Who is the error computed for?**
> *   To enable direct comparison with methods that don't consider protein flexibility, errors are computed for all heavy atoms of ligands.
>
> We have revised the manuscript to clarify these two evaluation contexts. We apologize for this mistake and thank you again for your careful correction!
>
>
>
> > **Q2:** The phrase "sampling every k = 10 steps" (Section 4.2.1) is imprecise without specifying the timestep used in the underlying simulations. The actual time interval (in ps or fs) should be stated explicitly.
>
> **A2:** We thank the reviewer for this important suggestion. We have updated the manuscript to provide these explicit values for all sampling procedures in Appendix  A.4.1.
>
> The sampling parameter `k=10` was used for three different datasets, which had different underlying simulation timesteps:
>
> - For the MISATO dataset, the all-atom MD simulations were performed with an 80 ps timestep. Therefore, our sampling of every `k=10` steps corresponds to a physical time interval of **0.8 ns** (`10 steps × 0.08ns /step`).
>
> - For the DD13M dataset, the simulations utilized a 0.1 ps (100 fs) timestep. Consequently, sampling every `k=10` steps resulted in a time interval of **1 ps** between saved frames (`10 steps × 0.1 ps/step`).
>
> -  In this revision, we also conducted new experiments on the ATLAS dataset, which contains 100 ns simulations for 1390 protein chains, allowing us to test BioMD's ability to capture larger conformational changes over longer timescales (Please go to **Common Response 1** for details about experiments on ATLAS). For the ATLAS dataset, each trajectory has a total simulation length of 100 ns and is provided as a sequence of 10,000 frames. We downsampled 10,000 frames to 1,000 frames in the preprocessing stage. Consequently, sampling every k=10 steps resulted in a time interval of **1 ns** between saved frames.

---

### Official Review · Reviewer_bQYZ · 2025-11-01

**Soundness:** 3
**Presentation:** 3
**Contribution:** 3
**Rating:** 6
**Confidence:** 3

**Summary:**

The paper proposes BioMD, a model for all-atom molecular dynamics prediction focusing on protein–ligand complexes. BioMD is based on flow matching conditioned on the protein sequence, ligand atom types, and the initial complex structure. By combining coarse-grained prediction for long time intervals with fine-grained interpolation for the short intervals in between, BioMD can model long molecular dynamics sequences. In the model architecture, the decoupling of intra-frame spatial attention and inter-frame temporal dependencies enables the simultaneous prediction of velocities across multiple time frames at all-atom resolution. To enhance physical plausibility, auxiliary losses are introduced. Experimental results demonstrate superior physical stability, conformational flexibility, and ligand unbinding success rates with BioMD.

**Strengths:**

1. The paper addresses an important problem: long-timescale molecular modeling of protein–ligand binding complexes at all-atom resolution, which has significant applications in drug discovery.
2. The efficient training and inference pipeline, along with the model architecture design, enables the prediction of multiple time frames simultaneously—crucial for modeling long MD trajectories. Experimental results show that the forecasting–interpolation workflow effectively reduces error accumulation.
3. The evaluation of the unbinding success rate demonstrates the model’s capability to capture real biomolecular processes and the authenticity of the predicted MD trajectories.
4. The paper is clearly written and well-presented.

**Weaknesses:**

1. Some implementation details are missing from both the main text and the appendix. For example, the architecture of the FlowTrajectoryTransformer is not fully described. In Algorithm 6, TemporalAttention and AttentionPairBias are used, but their definitions are missing, and only limited information about these modules is provided in the main text.
2. The evaluation focuses primarily on low-dimensional metrics, including bond/angle geometry and RMSF. The unbinding process is described using the centroid coordinates of the ligand. These metrics can reflect trajectory quality to some extent, but higher-dimensional evaluations or demonstrations would be more informative. For instance, analyzing distributional similarity between generated and ground-truth trajectories projected onto two TICA components, or examining the transition processes between clusters of conformers in the projected space, would provide deeper insight.

**Questions:**

1. In the formulation of the velocity model in Equation (2) (line 195), $\mathbf{Z}$ is said to contain the “amino acid sequence” and “ligand atom types.” Since the model is all-atom, how and when is atom-wise information from the residues incorporated into the model? From Figure 3, it seems that the atom-wise information of residues is only used in the velocity module after the $SE(3)$ Transformer backbone.
2. The paper focuses on protein–ligand complexes. However, the proposed techniques could, in principle, be applied to protein monomers and multimers as well. What is the motivation for focusing solely on protein–ligand complexes in this work?
3. How is the time interval in the coarse-grained forecasting stage chosen? How does the scale of this interval influence the quality and efficiency of trajectory generation?

---

> ### Author Response · Authors · 2025-11-26
>
> We sincerely thank Reviewer bQYZ for the constructive feedback and attention to detail. In response to your comments, we have significantly supplemented the manuscript with the missing implementation details, including pseudocode for attention modules and a full hyperparameter table. Furthermore, following your excellent suggestion, we have incorporated a higher-dimensional TICA analysis on the ATLAS dataset to demonstrate our model's superior conformational sampling capabilities compared to short MD simulations. Our point-by-point responses are listed below.
>
> > **Weakness 1:** Some implementation details are missing from both the main text and the appendix. For example, the architecture of the FlowTrajectoryTransformer is not fully described. In Algorithm 6, TemporalAttention and AttentionPairBias are used, but their definitions are missing, and only limited information about these modules is provided in the main text.
>
> **A1:** We thank the reviewer for pointing out this lack of clarity. To address this, we add explicit definitions for the `TemporalAttention` and `AttentionPairBias` modules in **Appendix A.1** and pseudocode in **Algorithm 7** and **Algorithm 8** respectively, clarifying how they operate on the temporal and spatial dimensions of the data. Further, a table with key hyperparameters of BioMD, including the number of layers, attention heads, and the dimensions of the single (`c_s`) and pair (`c_z`) representations, is also added in **Appendix A.4.2**.
>
>
>
> > **Weakness 2:** The evaluation focuses primarily on low-dimensional metrics... These metrics can reflect trajectory quality to some extent, but higher-dimensional evaluations or demonstrations would be more informative. For instance, analyzing distributional similarity between generated and ground-truth trajectories projected onto two TICA components... would provide deeper insight.
>
> **A2:** We thank the reviewer for this excellent suggestion! We agree that a higher-dimensional evaluation is essential to provide deeper insight into model performance than scalar metrics alone.
>
> To address your point, we first conducted experiments on the ATLAS dataset (see **Common Response 1**). For proteins in the ATLAS test set, we performed a **Time-lagged Independent Component Analysis (TICA)**. The first two independent components (tIC1 and tIC2), which capture the slowest and most significant motions of the system, were used to define a 2D conformational space. We first fit TICA on a full 100 ns Molecular Dynamics (MD) trajectory. We then projected the conformations generated by our model, **BioMD**, onto this 2D space. For comparison, we also projected conformations from other MD simulations (100ns, 10ns, and 1ns) onto the same space.
>
> The results in Figure 13 in the appendix clearly demonstrate the strength of our approach:
>
> *   The results show that short MD simulations (**MD 1ns** and **MD 10ns**) are severely limited in their exploration. They typically only sample a very small, localized region of the total conformational space, often missing other important low-energy states entirely.
>
> *   In stark contrast,  **BioMD** generates a diverse ensemble of structures that effectively covers the key low-energy basins seen in the 100ns ground-truth trajectory. For some proteins with multiple distinct states (e.g., **6H86, 6L4P, 7BWF**), BioMD successfully samples conformations in each of the major basins.
>
> Notably, the average sampling time for one trajectory is only 56 seconds, while a 10ns protein MD simulation requires approximately one hour of computation in a single GPU, yet usually fails to adequately explore the conformational space. Therefore, this comparison proves BioMD reproduces the equilibrium conformational distribution, substantially outperforming short MD in both sampling quality and speed.

---

> > ### Author Response · Authors · 2025-11-26
> > **Response to questions**
> >
> > > **Q1:** In the formulation of the velocity model in Equation (2) (line 195), $mathbf{Z}$ is said to contain the “amino acid sequence” and “ligand atom types.” Since the model is all-atom, how and when is atom-wise information from the residues incorporated into the model? From Figure 3, it seems that the atom-wise information of residues is only used in the velocity module after the Transformer backbone.
> >
> > **A1:**  We thank the reviewer for this question about clarity. We clarify that atom-wise information is incorporated **at the initialization stage**, meaning the Transformer backbone operates entirely on all-atom representations.
> >
> > Although $\mathbf{Z}$ contains the "amino acid sequence," we do not feed the sequence directly into the network. Instead, we first expand the sequence into a set of atoms using standard chemical topology lookups. These derived atom features (along with ligand atoms) form the initial single representation $\mathbf{s}_i$, which serves as the input to the `FlowTrajectoryTransformer`. Therefore, the Transformer processes atom-level features throughout all its layers. The final velocity module mentioned in the question is merely a projection head that converts these processed atom embeddings into velocity vectors. We have clarified this in our revised paper. Thanks again for your careful review!
> >
> >
> > > **Q2:** The paper focuses on protein–ligand complexes. However, the proposed techniques could, in principle, be applied to protein monomers and multimers as well. What is the motivation for focusing solely on protein–ligand complexes in this work?
> >
> > **A2:** Thank you for this insightful question! You are absolutely correct that our proposed framework is general. To directly validate its applicability to systems beyond protein-ligand complexes, we have expanded our evaluation to include protein monomers.
> >
> > Specifically, we conducted new experiments on the ATLAS dataset [1], which contains 100 ns simulations for 1390 protein chains, allowing us to test BioMD's ability to capture larger conformational changes over longer timescales. Please go to **Common Response 1** for a detailed response.
> >
> >
> > > **Q3:** How is the time interval in the coarse-grained forecasting stage (or "large-step" size) chosen? How does the scale of this interval influence the quality and efficiency of trajectory generation?
> >
> > **A3:** We thank the reviewer for this important question regarding our hyperparameter selection. Our choice of an autoregressive block size of 5 (AR-5) and a hierarchical step size of 10 (k=10) is empirically driven, primarily based on the performance for both tasks, and we have clarified this in **Appendix A.5** of our revised manuscript. Please go to **Common Response 2** for a detailed response.

---

> ### Comment · Reviewer_bQYZ · 2025-11-27
>
> Thanks for the response. The additional results on longer protein dynamics shows better generalizability of BioMD. And thanks for providing the additional details of the model architecture. I will raise my score to 8.

---

> ### Author Response · Authors · 2025-11-27
>
> Thank you very much for your prompt reply and raising the score to 8! Your thoughtful suggestions mean a lot to us. If you have any further questions, please feel free to reach out to us. Once again, we all sincerely appreciate your time and effort.

---

### Official Review · Reviewer_1cTN · 2025-11-01

**Soundness:** 2
**Presentation:** 3
**Contribution:** 2
**Rating:** 4
**Confidence:** 2

**Summary:**

This paper introduces BioMD, a novel all-atom generative model designed to simulate long-timescale protein-ligand systems. The primary motivation is to overcome the significant computational cost of traditional Molecular Dynamics simulations, which severely limits the accessible timescales for many biologically relevant processes. The core contribution is a hierarchical framework that decomposes the complex task of long-trajectory generation into coarse-grained forecasting and fine-grained interpolation. The authors validate BioMD on two datasets. BioMD generates trajectories with high physical plausibility and successfully generates complete ligand unbinding paths.

**Strengths:**

1. The hierarchical  framework decomposes the long-sequence problem into "capturing the backbone first, then filling the details,"  effectively reduces the sequence length and the complexity of long-range dependencies in the generation process.
2. Using a single flow matching model to achieve two different tasks (forecasting and interpolation) simply by changing the masking schedules.
3. Supportive experimental results, especially the ligand unbinding experiment.
4. The paper is clearly written and presented.

**Weaknesses:**

1. **Definition of "Long-Timescale"**: The paper claims to simulate "long-timescale" dynamics. However, the MISATO dataset's simulation time (8 ns) is relatively short, insufficient to observe major protein conformational transitions (e.g., open-to-closed states). While DD-13M is a dissociation task, it was generated by metadynamics, an enhanced sampling method, not a spontaneous process in equilibrium. Therefore, the model's ability to simulate equilibrium rare events at the microsecond (µs) to millisecond (ms) level has not been demonstrated
2. **Insufficient Evaluation of Protein Conformational Changes**: The evaluation of protein flexibility relies mainly on RMSF, which is an aggregate metric. In drug discovery, we are often more concerned with specific conformational changes near the binding pocket, such as the flipping of key side-chains or the movement of "gating" residues. The paper lacks an in-depth case study on these critical biological events.
3. The model's performance on DD-13M is highly dependent on the quality and coverage of the training data. The DD-13M dataset was generated using metadynamics, which means BioMD may have learned a biased distribution of "how to dissociate quickly" rather than the dynamics of the true Boltzmann distribution. Could the author provide some study on the OOD cases using BioMD?
4. Lack of ablation studies, such as how the definition of "large-step" influences the final results and what is the effect of different sampling steps.

**Questions:**

- Was any data cleaning performed on the MISATO dataset? For instance, were unstable simulation results (according to the RMSD result) filtered out before training?

 - Did you evaluate the difference between the generated protein conformations and the reference MD simulations, particularly regarding key conformational changes in the binding pocket?

- Have you attempted to extend the model to even longer timescales (beyond those in DD-13M)? Given the acknowledged error accumulation, do you observe simulation collapse or instability over extended periods? Furthermore, the MISATO simulations are relatively short (8 ns), making it difficult to observe significant events like major conformational transitions. Could you provide representative case studies to demonstrate the model's ability (or limitations) in capturing such rare but significant events?

- Regarding Figure 6, how consistent are the generated unbinding pathways with the reference metadynamics results?

- I recommend including related works that do not directly recover the dynamics trajectories but recover the key binding pose and also use flow matching or diffusion models. There are some related works recommended. For example, DynamicBind [1] learns to transform unbound to bound states of protein-ligand complexes; DynamicFlow [2] uses a flow model to transform protein apo state and a noisy ligand to a holo state and a binding ligand to consider protein dynamics in structure-based drug design.


**Note: I might raise my score according to the situation of rebuttal discussion and paper revision.**

References:

[1] DynamicBind: predicting ligand-specific protein-ligand complex structure with a deep equivariant generative model, Nature Machine Intelligence 2024

[2] Integrating Protein Dynamics into Structure-Based Drug Design via Full-Atom Stochastic Flows. ICLR 2025.

**Details Of Ethics Concerns:**

None.

---

> ### Author Response · Authors · 2025-11-26
>
> We thank the reviewer for the valuable comments and constructive feedback. In response, we have significantly strengthened our evaluation by incorporating the ATLAS dataset for longer-timescale validation and conducting detailed analyses of binding pocket dynamics (side-chains and contact maps). Furthermore, we have added the requested ablation studies, OOD generalization cases, and expanded our discussion on related works and model limitations. Our point-by-point responses are listed below.
>
> > **Weakness 1:** Definition of "Long-Timescale": The paper claims to simulate "long-timescale" dynamics ....... the model's ability to simulate equilibrium rare events at the microsecond (µs) to millisecond (ms) level has not been demonstrated.
>
> **A1:** Thank you for this crucial point! To more directly address your concern, we have expanded our evaluation to the ATLAS dataset [1], which contains 100 ns simulations for 1390 protein chains, allowing us to test BioMD's ability to capture larger conformational changes over longer timescales. Please go to **Common Response 1** for a detailed response.
>
> Finally, we acknowledge the limitation of BioMD on simulating rare events due to the current lack of MD simulation data of the microsecond (µs) to millisecond (ms) timescale. To this end, our future work will focus on collecting extensive long-timescale MD simulation data, which we will use to fine-tune our model to enhance its predictive capabilities for these crucial, slower dynamic events.
>
>
>
> > **Weakness 2 & Question 2:** Insufficient Evaluation of Protein Conformational Changes: The evaluation of protein flexibility relies mainly on RMSF ...... particularly regarding key conformational changes in the binding pocket?
>
> **A2:** We thank the reviewer for this excellent suggestion! To address this, we perform two new in-depth analyses focusing on the specific, functionally important conformational changes near the protein-ligand interface.
>
> 1.  **Side-Chain Dynamics (Dihedral Distributions):** We analyzed the dihedral angle distributions for side-chains closest to the ligand. As shown in **Figure 9**, the distributions generated by BioMD (dashed red) show strong agreement with the ground truth MD simulations (solid blue). This demonstrates that our model can accurately capture the flipping of key side-chains, which are critical for ligand binding and recognition.
>
> 2.  **Protein-Ligand Interactions (Contact Maps):** We computed and compared protein-ligand contact maps to evaluate the stability of the binding interface. **Figure 10** shows that the contact probability heatmaps generated from BioMD trajectories are highly similar to those from the ground truth MD simulations. This confirms that our model correctly preserves the network of critical interactions that define the binding pose and pocket environment.
>
> Together, these focused analyses provide strong evidence that BioMD successfully models the functionally critical dynamics within the binding pocket, going well beyond what aggregate metrics like RMSF can show.
>
>
>
>
>
> > **Weakness 3:** The model's performance on DD-13M is highly dependent on the quality and coverage of the training data ...... how consistent are the generated unbinding pathways with the reference metadynamics results?
>
> **A3:** Thank you for this insightful point regarding the nature of the learned distribution. We agree that by training on metadynamics data, BioMD learns a model of an "accelerated" unbinding process, not the true thermodynamic or kinetic one. We have clarified this in the **Section 5.2** of the revised paper.
>
> To address your questions about robustness on Out-of-Distribution (OOD) cases and pathway consistency, we have performed two key analyses:
>
> 1.  **OOD Generalization Case Study:** We present an OOD case study in **Figure 12** on the DD-13M protein (PDB ID: 3PCU), which has a low sequence similarity (0.26) to any protein in our training set. The visualization shows that BioMD generates a diverse set of unbinding poses that explore the same exit channels (Clusters A, B, C) as the ground-truth metadynamics simulation. For this challenging OOD target, our model still achieves a high **correct rate of 0.92** for pathways coming within 2 Å of the true exit, demonstrating its generalization capability.
>
> 2.  **Quantitative Pathway Consistency:** To provide a more rigorous evaluation of whether our model "finds the correct way out" across the test set, we introduced two new metrics (**Appendix 3.5**). Please go to **Common Response 5** for a detailed response.
>
>
> > **Weakness 4:** Lack of ablation studies, such as how the definition of "large-step" influences the final results and what is the effect of different sampling steps.
>
> **A4:** We thank the reviewer for this important question regarding ablation studies. We have conducted these studies, and the results are presented in **Tables 7, 8** of our manuscript. Please go to **Common Response 2** for a detailed response.

---

> > ### Author Response · Authors · 2025-11-26
> > **Responses to questions**
> >
> > > **Q1: Was any data cleaning performed on the MISATO dataset? For instance, were unstable simulation results (according to the RMSD result) filtered out before training?**
> >
> > Yes, but minimal data cleaning was performed. We only filter out complexes in MISATO with sequence length longer than 800 and ligand heavy atoms greater than 100. We have checked the data, and there are some unstable simulations of very weakly binding samples, and we did not exclude them from training for the purpose of learning real dynamics. We have clarified this in **Section 5.1**.
> >
> >
> >
> > > **Q3: Have you attempted to extend the model to even longer timescales (beyond those in DD-13M)? Given the acknowledged error accumulation, do you observe simulation collapse or instability over extended periods? Furthermore, the MISATO simulations are relatively short (8 ns), making it difficult to observe significant events like major conformational transitions. Could you provide representative case studies to demonstrate the model's ability (or limitations) in capturing such rare but significant events?**
> >
> >
> > Thank you for your question about generalization on longer timescales. We conducted experiments on generating 1,000 frames in the DD-13M test set (500 frames for previous experiments, an average of 480 frames for the DD-13M dataset). We find that extending inference steps on DD-13M yielded a marginal improvement in unbinding success. However, this is accompanied by significant error accumulation problems, leading to a **2254 times** increase in protein-ligand clash score. We have added this discussion in the ablation study part of our revised paper (**Appendix A.5** and **Table 8**).
> >
> > For a case study on major conformational transitions, we trained BioMD on the ATLAS dataset and provided a case study for domain motion on the Adenylate Kinase (see response for **Weakness 1** and **Figure 6** in the revised paper).
> >
> >
> >
> >
> > > **Q4: Regarding Figure 6, how consistent are the generated unbinding pathways with the reference metadynamics results?**
> >
> > Thank you for pointing out this question! We have noted the unbinding path RMSD in both **Figure 5** and our newly added OOD case study **Figure 12**.
> >
> >
> >
> > > **Q5: I recommend including related works that do not directly recover the dynamics trajectories but recover the key binding pose and also use flow matching or diffusion models... DynamicBind [1] and DynamicFlow [2].**
> >
> > Thank you for pointing out these important works! We have cited and discussed DynamicBind and DynamicFlow in the **Related Works** section. Specifically, we clarified that they focus on generating binding poses with protein flexibility, while our work targets kinetic trajectory generation.

---

### Official Review · Reviewer_TABm · 2025-11-02

**Soundness:** 3
**Presentation:** 3
**Contribution:** 3
**Rating:** 6
**Confidence:** 3

**Summary:**

The paper introduces BioMD, an all-atom generative model for biomolecular dynamics simulation. It combines flow matching with a hierarchical forecasting–interpolation framework to generate long-timescale protein–ligand trajectories efficiently. The model is evaluated on MISATO and DD-13M, achieving high physical plausibility and impressive ligand-unbinding success rates.

**Strengths:**

* The work addresses a challenging and underexplored problem: generating full, time-resolved protein–ligand trajectories at all-atom resolution. The motivation is clearly articulated and relevant to molecular simulation and drug discovery.

* The proposed hierarchical framework is conceptually elegant. Separating long-range forecasting from local interpolation aligns well with physical intuition and effectively mitigates error accumulation over long trajectories.

* The unified conditional flow-matching formulation (“noising-as-masking”) is technically interesting and flexible, enabling both forecasting and interpolation within one architecture.

* The experiments are comprehensive within the chosen scope.

* The auxiliary geometric losses (bond, collision, and center penalties) are physically meaningful and help ensure chemically plausible outputs.

**Weaknesses:**

* The paper mainly reports geometric errors and RMSF correlations, but omits more direct physical checks (e.g., energy or temperature conservation, relaxation consistency). Without such validation, “physical plausibility” remains largely geometric.

* The experimental baselines are narrow. Although the authors mention Str2Str (Lu et al., 2024) and BioEmu (Lewis et al., 2025) in the related work, these strong generative baselines are not included in the experiments. The justification that they are “time-agnostic” is somewhat artificial, since BioMD also measures static geometric stability and RMSF correlations—metrics directly comparable to those ensemble models. Moreover, both Str2Str and BioEmu are flow- or diffusion-based architectures closely related to BioMD’s conditional flow-matching design. Their absence weakens the empirical comparison and makes it unclear whether BioMD truly surpasses existing conformational generators.

*  The architecture figure is dense but lacks quantitative configuration (model size, training schedule, runtime benchmarking).

* It is unclear whether the hierarchical masking approach can generalize to unseen complexes or trajectories longer than those used in training. A discussion of scalability and failure cases would strengthen the work.

**Questions:**

See above

---

> ### Author Response · Authors · 2025-11-26
>
> We thank the reviewer for the insightful and constructive comments. To address your concerns, we have added energy-based relaxation experiments to verify physical plausibility and included comparisons with Str2Str and BioEmu on the ATLAS dataset. Additionally, we have refined the architecture details and expanded the discussion on generalization and scalability. Our point-by-point responses are listed below.
>
> > **W1:** The paper mainly reports geometric errors and RMSF correlations, but omits more direct physical checks (e.g., energy or temperature conservation, relaxation consistency). Without such validation, “physical plausibility” remains largely geometric.
>
> **A1:** Thank you for your insightful suggestion! We have now performed an additional analysis to assess the energetic quality of our generated conformations.
>
> Specifically, we conducted a **structural relaxation experiment**. For each trajectory generated by BioMD on the MISATO test set, we took each frame and performed energy minimization using the Amber99SB force field. We then calculated the all-atom RMSD between the pre- and post-relaxation structures. A low RMSD indicates that our generated conformations are already in or very near a local energy minimum, confirming their physical plausibility.
>
> Our analysis reveals a mean all-atom RMSD of **0.72 Å** (**Figure 8**) after relaxation across all generated frames (**0.69 Å** for MD conformations, **Figure 8**), demonstrating the ability of BioMD to generate energetically favorable structures and providing strong evidence for the physical plausibility of our generated trajectories beyond purely geometric metrics.
>
>
> > **W2:** The experimental baselines are narrow. Although the authors mention Str2Str (Lu et al., 2024) and BioEmu (Lewis et al., 2025) in the related work, these strong generative baselines are not included in the experiments. Their absence weakens the empirical comparison.
>
> **A2:** Thank you for your constructive suggestion! We have expanded our experimental section to include these strong baselines. As Str2Str and BioEmu are specially designed for protein systems, we conducted a new set of experiments on the ATLAS dataset, comprising 100 ns MD simulations for 1390 protein chains. Please go to **Common Response 1** for a detailed response.
>
>
> > **W3:** The architecture figure is dense but lacks quantitative configuration.
>
> **A3:** We thank the reviewer for this valuable and constructive feedback! To address this, we have revised the architecture diagram in **Figure 3** to improve clarity and incorporate key configuration parameters directly. Specifically, we have annotated the diagram with:
>
> *   The total number of model parameters (341M).
>
> *   The configuration of the core SE(3)-graph transformer, which has 6 layers and 16 heads.
>
> *   The number of heads used in the sequential local attention blocks (3 blocks and 4 heads) and global attention blocks (24 blocks and 16 heads).
>
> Besides, we have also added all key hyperparameters in **Appendix A.4.2** and **Table 4,5,6** for clarity. Thanks again for the suggestion.
>
>
> > **W4:** It is unclear whether the hierarchical masking approach can generalize to unseen complexes or trajectories longer than those used in training. A discussion of scalability and failure cases would strengthen the work.
>
> **A4:** Thank you for raising this critical and insightful point! We have substantially revised the manuscript to address these aspects more explicitly.
>
> **1. Generalization to Unseen Complexes:**
>
> To probe this in a challenging Out-of-Distribution (OOD) scenario, we analyzed a complex (PDB: 3PCU), whose largest sequence similarity with the training set is 0.26. As shown in **Figure 12** of the revised version, our model successfully identified the same unbinding pathways as the ground truth Metadynamics simulation, demonstrating its ability to predict physically plausible dynamics for novel systems.
>
> **2. Generalization and Scalability to Longer Trajectories:**
>
> We conducted experiments on generating 1,000-length trajectories for the DD-13M test set (500 frames for previous experiments, an average of 480 frames for the DD-13M dataset). We find that extending inference steps yielded a marginal improvement in unbinding success. However, this is accompanied by significant error accumulation problems, leading to a significant degradation in physical stability metrics (2254 times increase in protein-ligand clash score). We have added this discussion in the ablation study part of our revised paper (**Appendix A.5** and **Table 8**).
>
> As for generating trajectories significantly longer than those used in training (e.g., at the microsecond (µs) to millisecond (ms) level), we acknowledge the limitation of BioMD on this task due to the lack of MD simulation data at these timescales. In future work, we aim to collect extensive long-timescale MD data to fine-tune our model and improve its predictive capabilities for these slower, yet crucial, dynamic events.

---

### Author Response · Authors · 2025-11-26
**Response to all reviewers**

We sincerely thank all the reviewers for their valuable feedback and insightful evaluation of our work. In response to the reviewers’ feedback, we have conducted new analyses and made corresponding changes to the paper to clarify our method and findings. In this revision, we have added **7 new figures and 7 new tables** and made substantial improvements to the text.

We believe our response has addressed all concerns raised, fully demonstrating the effectiveness of BioMD in generating physically plausible and kinetically accurate biomolecular trajectories. Among all reviewers, we have received a response from **Reviewer bQYZ**. We are particularly grateful to Reviewer bQYZ for the prompt feedback and for recognizing the value of our work by raising the score to **8**.

We have categorized the reviewers' questions into 6 main areas. Our major improvements include:

**1. New experiments on the ATLAS dataset (Raised by: TABm, 1cTN, bQYZ)**
To address concerns regarding the evaluation of long-timescale dynamics and comparison with strong generative baselines, we expanded our evaluation to the ATLAS dataset (100 ns simulations), comparing BioMD against strong baselines like Str2Str and BioEmu. BioMD achieves state-of-the-art performance on 9 out of 13 metrics (**Table 3**), successfully modeling the domain motion of Adenylate Kinase (**Figure 6**) and demonstrating superior conformational space exploration via TICA analysis (**Figure 13**).

**2. Rigorous Ablation Studies (Raised by: 1cTN, bQYZ, gM8W)**
We conducted extensive ablation studies on both MISATO and DD-13M datasets to justify the hierarchical architecture and hyperparameter choices (step size $k$ and autoregressive block size), while also analyzing error distributions. The results validate the necessity of our hierarchical architecture and specific parameter choices (**Tables 7, 8**).

**3. Detailed Physical Plausibility Evaluation (Raised by: TABm, gM8W)**
We performed structural relaxation experiments using energy minimization to verify that generated structures represent local energy minima. The resulting boxen plots visualizing per-frame error distributions (**Figure 7**) and histogram plots showing low RMSD between generated and relaxed structures (**Figure 8**) confirm the energetic stability and physical plausibility of BioMD's outputs.

**4. Detailed Dynamics Analysis (Raised by: 1cTN, bQYZ)**
We moved beyond aggregate metrics (RMSF) to analyze specific binding pocket dynamics and high-dimensional conformational space. We demonstrated accurate modeling of side-chain dihedral distributions (**Figure 9**) and protein-ligand contact maps (**Figure 10**), while TICA analysis confirms BioMD captures diverse conformational basins better than short MD (**Figure 13**).

**5. New Unbinding Metrics (Raised by: 1cTN, gM8W)**
To rigorously evaluate whether BioMD identifies the correct unbinding pathway rather than simply finding "any" exit, we introduced *Endpoint Distance* and *Unbinding Correct Rate*. BioMD achieves a high correct rate consistent with ground truth (**Table 9**), proving it learns valid unbinding pathways.

**6. Others:**
*   New OOD Case Study on unseen protein (**Figure 12**).
*   Clarification of architecture details (**Figure 3**) and hyperparameter settings (**Tables 4,5,6**).

Below, we provide detailed common responses for specific categories and individual responses to each reviewer.

---

> ### Author Response · Authors · 2025-11-26
> **Common Response 1: New Experiments on the ATLAS Dataset**
>
> To address concerns regarding the evaluation of long-timescale dynamics and comparison with strong generative baselines (Reviewers TABm, 1cTN, bQYZ), we have expanded our evaluation to the **ATLAS dataset** [1]. This dataset contains 100 ns MD simulations for 1390 protein chains, allowing us to test BioMD's ability to capture larger conformational changes over longer timescales compared to the MISATO dataset (8 ns).
>
> We compared BioMD against state-of-the-art generative models, including **Str2Str** and **BioEmu**. As shown in **Table 3**, BioMD achieves state-of-the-art performance on 9 out of 13 metrics for conformational ensemble generation. Specifically:
>
> *   **Conformational Transitions:** BioMD successfully captures major protein conformational transitions, and we conducted a case study to model the domain motion (between open and closed states) of Adenylate Kinase (**Figure 6**). Starting from the closed state (1AKE, 0 ns), BioMD simulates the opening process, reaching a conformation at 23 ns that closely aligns with the experimental open state (4AKE) with an RMSD of 1.631 Å. This indicates that our model can capture major protein conformational transitions that are biologically significant.
> *   **Space Exploration:** Time-lagged Independent Component Analysis (TICA) in **Figure 13** and **Appendix A.6.4** demonstrates that BioMD explores the conformational space more effectively than short-timescale MD simulations (1 ns and 10 ns), capturing a diversity closer to the full 100 ns ground truth trajectories. Notably, the average sampling time for one trajectory is only 56 seconds, while a 10ns protein MD simulation requires approximately one hour of computation in a single GPU, yet usually fails to adequately explore the conformational space.
>
> [1] Vander Meersche, Y., et al. (2023). ATLAS: protein flexibility description from atomistic molecular dynamics simulations. Nucleic Acids Research.
>
> **Table 3: Results on the ATLAS test set. Performance comparison of methods for modeling protein dynamics across different metrics.**
>
> |Category|Metrics|ESMFLow-MD Full|ESMFLow-MD Distilled|ConfDiff|BioEmu|Str2Str|MDGen|EBA|BioMD|
> |---|---|:-:|:-:|:-:|:-:|:-:|:-:|:-:|:-:|
> |**Predicting flexibility**|Pairwise RMSD r ↑|0.19|0.19|0.59|0.46|0.23|0.48|0.62|**0.70**|
> ||Global RMSF r ↑|0.31|0.33|0.67|0.57|0.38|0.50|0.71|**0.76**|
> ||Per-target RMSF r ↑|0.76|0.74|0.85|0.71|0.57|0.71|0.90|**0.91**|
> |**Distributional accuracy**|Root mean W₂-dist. ↓|3.60|4.23|2.76|4.32|4.05|2.69|2.43|**2.18**|
> ||↪ Trans. contrib. ↓|3.13|3.75|2.23|4.04|3.73|-|2.03|**1.89**|
> ||↪ Var. contrib. ↓|1.74|1.90|1.40|1.77|1.43|-|1.20|**1.10**|
> ||MD PCA W₂-dist. ↓|1.51|1.87|1.44|1.97|2.04|1.89|**1.19**|1.24|
> ||Joint PCA W₂-dist. ↓|3.19|3.79|2.25|3.98|3.55|-|2.04|**1.82**|
> ||% PC-sim > 0.5 ↑|26|33|35|**51**|12|-|44|46|
> |**Ensemble observables**|Weak contacts J ↑|0.55|0.48|0.59|0.33|0.43|0.51|**0.65**|0.61|
> ||Transient contacts J ↑|0.34|0.30|0.36|-|-|-|0.41|**0.46**|
> ||Exposed residue J ↑|0.49|0.43|0.50|-|-|0.29|0.70|**0.70**|
> ||Exposed MI matrix ρ ↑|0.20|0.16|0.24|0.07|0.21|-|**0.36**|0.34|

---

> ### Author Response · Authors · 2025-11-26
> **Common Response 2: Ablation Studies on the MISATO and DD-13M Dataset**
>
> In response to questions about the necessity of our hierarchical architecture, hyperparameter choices, and lack of ablation study (Reviewers 1cTN, bQYZ, gM8W), we conducted extensive ablation studies on both MISATO and DD-13M datasets (**Table 7 & 8**, **Appendix A.5**). We analyzed the impact of the hierarchical step size ($k$) and the autoregressive block size ($AR$).
>
> Our key findings justify our design choices:
>
> 1.  **The choice of AR-5 and k=10 is optimal for unbinding.** As shown in our ablation studies, this specific combination yields the best results across the two primary metrics for the unbinding task.
>
> 2.  **The model is highly sensitive to the block size (AR-x).**
>     *   A small block  (AR-1) requires too many sequential inference steps, leading to severe error accumulation. This results in significant physical instability, with protein-ligand clashes increasing to over **4000 times** the baseline.
>
>     *   A large block  (AR-10 and w.o. AR) appears to increase the task's difficulty, potentially confusing the model and degrading performance, especially on unbinding metrics (Unbinding Path RMSD and Unbinding Success Rate).
>
> 3.  **Hierarchical architecture is crucial for reducing error accumulation.** A non-hierarchical model (`k=1`) suffers from extreme error accumulation due to the vast number of inference steps needed. This also results in significant physical instability, with protein-ligand clashes reaching over **7000 times** the baseline.
>
> **Table 7: Ablation study on the MISATO dataset.** Comparison of all methods on physical stability and conformational flexibility.
>
> |Method|Bond MAE|Bond MSE|Angle MAE|Angle MSE|Clash(Intra Lig)|Clash(Prot-Lig)|RMSF Corr Ligand|RMSF Corr Protein|RMSF Ligand (1.211)|RMSF Protein (1.002)|
> |---|---|---|---|---|---|---|---|---|---|---|
> |Molecular Dynamics|.0377|.0023|.0575|.0053|0|0|-|-|-|-|
> |BioMD (k=1)|**.0356**|**.0021**|**.0568**|**.0053**|0|0|.4526|.6291|.3717|.3825|
> |BioMD (k=5)|.0387|.0024|.0627|.0066|0|.0001|**.4982**|.6023|.4552|.4287|
> |BioMD (k=10)|.0495|.0155|.0709|.0097|.0019|.0023|.4789|**.6854**|**.7023**|**.6242**|
> |BioMD (k=20)|.0421|.0030|.0698|.0087|.0002|.0008|.4267|.5309|.5403|.4959|
>
> **Notes:** *k=1 represents the "forecasting-only" version of BioMD.*
>
>
>
> **Table 8: Ablation study on the DD-13M dataset.** Comparison of methods on physical stability, ligand unbinding path reconstruction metric, and ligand unbinding success rates.
>
> |Method|Bond MAE|Bond MSE|Angle MAE|Angle MSE|Clash(Intra)|Clash(Prot-Lig)|Path RMSD|Succ@1|Succ@5|Succ@10|
> |---|---|---|---|---|---|---|---|---|---|---|
> |Static|-|-|-|-|.2778|.0000|.6504|-|-|-|
> |Metadynamics|.0246|.0012|.0452|.0030|.2777|.0000|.4217|-|-|-|
> |BioMD (AR-1, k=10)|.1880|.0612|.2011|.0761|1.727|4.079|1.131|.2285|.4728|.5822|
> |BioMD (AR-5, k=10)|.0728|.0111|.0802|.0132|**.2943**|.0009|**.5645**|.5676|.7419|.7941|
> |BioMD (AR-10, k=10)|.1067|.0221|.1478|.0450|.7098|.5356|.7465|.4008|.5719|.6239|
> |BioMD (w.o. AR, k=10)|**.0308**|**.0018**|**.0606**|**.0077**|.2944|**.0004**|.6845|.0029|.0147|.0294|
> |BioMD (AR-5, k=10, L\*2)|.1457|.0382|.1770|.0595|1.407|2.029|.7766|**.5735**|**.7512**|**.8010**|
> |---|||||||||||
> |BioMD (AR-5, k=1)|.0682|.0179|.1231|.0321|.3686|7.380|.7313|.3428|.5128|.6293|
> |BioMD (AR-5, k=5)|.0575|.0164|.1062|.0246|.4700|1.346|.5703|.5214|.6582|.7023|
> |BioMD (AR-5, k=10)|.0728|.0111|.0802|.0132|**.2943**|**.0009**|**.5645**|**.5676**|**.7419**|.7941|
> |BioMD (AR-5, k=20)|**.0497**|**.0053**|**.0914**|**.0184**|.3736|.0967|.5909|.4714|.7065|**.8043**|
>
> **Notes:** *(1). k=1 represents the "forecasting-only" version of BioMD; (2). "L\*2" means that the generated trajectory length is doubled (i.e., 1000 frames).*

---

> ### Author Response · Authors · 2025-11-26
> **Common Response 3: Detailed Physical Plausibility Evaluation**
>
> To address concerns regarding the energetic quality of our generated conformations (Reviewers TABm, gM8W), we have performed additional analyses to assess physical plausibility beyond geometric metrics.
>
> **Structural Relaxation and Energy Minimization:**
> We conducted a structural relaxation experiment to verify that generated structures are close to local energy minima. For each trajectory generated by BioMD on the MISATO test set, we took every frame and performed energy minimization using the Amber99SB force field. We then calculated the all-atom RMSD between the pre- and post-relaxation structures. A low RMSD indicates that our generated conformations are near a local energy minimum, confirming their physical plausibility.
>
> **Results:**
> *   **Per-Frame Error Distributions (Figure 7, Appendix A.6.1):** The resulting boxen plots visualize the error distributions, demonstrating consistent stability across the generated trajectories.
> *   **Relaxation RMSD (Figure 8, Appendix  A.6.2 ):** Our analysis reveals a mean all-atom RMSD of **0.72 Å** after relaxation across all generated frames. This is highly comparable to the **0.69 Å** RMSD observed for ground truth MD conformations under the same relaxation protocol.
>
> These results demonstrate BioMD's ability to generate energetically favorable structures and provide strong evidence for the physical plausibility of our generated trajectories.

---

> ### Author Response · Authors · 2025-11-30
> **Common Response 4: Detailed Dynamics Analysis**
>
> In response to requests for more in-depth evaluation of protein conformational changes and high-dimensional dynamics (Reviewers 1cTN, bQYZ), we performed three new analyses focusing on binding pocket dynamics and conformational space exploration.
>
> **1. Side-Chain Dynamics (Dihedral Distributions):**
> We analyzed the dihedral angle distributions for side-chains closest to the ligand. As shown in **Figure 9** and **Appendix A.6.3**, the distributions generated by BioMD (dashed red) are close to the ground truth MD simulations (solid blue), demonstrating that BioMD can accurately capture the flipping of key side-chains, which are critical for ligand binding.
>
> **2. Protein-Ligand Interactions (Contact Maps):**
> We computed and compared protein-ligand contact maps to evaluate the accuracy of our predicted binding interface. **Figure 10** shows that the contact probability heatmaps generated from BioMD trajectories are highly similar to those from the ground truth MD simulations. This confirms that our model correctly simulates the critical interactions that define the binding pose and affinity.
>
> **3. High-Dimensional Space Exploration (TICA):**
> To evaluate the global dynamics of our generated trajectories, we performed Time-lagged Independent Component Analysis (TICA) on the ATLAS dataset. Details are discussed in **Common Response 1**.

---

> ### Author Response · Authors · 2025-11-30
> **Common Response 5: New Unbinding Metrics on Endpoint Distance and Unbinding**
>
> To rigorously evaluate whether BioMD identifies the *correct* unbinding pathway rather than simply finding *any* exit (Reviewers 1cTN, gM8W), we introduced two new metrics in **Appendix A.3.5**:
>
> *   **Endpoint Distance:** The minimum RMSD between any predicted ligand position along the path and the ground-truth exit position.
> *   **Correct Rate:** The percentage of trajectories where the Endpoint Distance is below a given threshold.
>
> The results in **Table 9** demonstrate that BioMD does not move randomly. Our best model achieves a **Correct Rate (< 1 Å) of 68.24%**, which closely approximates the reference Metadynamics simulations (75.32%) and significantly outperforms the static baseline. This confirms that BioMD effectively learns physically plausible unbinding pathways present in the training data and generates unbinding pathways that terminate at the correct exit region.
>
> **Table 9: Comparison of methods on endpoint distance and unbinding correct rates.**
> |Method|Endpoint Distance (RMSD, Å)|Correct Rate (< 0.5 Å)|Correct Rate (< 1 Å)|Correct Rate (< 2 Å)|Correct Rate (< 4 Å)|
> |---|---|---|---|---|---|
> |**Static**|3.5852|0.1765|0.3235|0.4118|0.7353|
> |**Metadynamics**¹|0.9739|0.5136|0.7532|0.9022|0.9641|
> |**BioMD-rel**|2.3433|0.4265|0.5265|0.6971|0.8000|
> |**BioMD-abs**|2.3897|0.3765|0.5324|0.6882|0.8059|
> |**BioMD-rel (AR-5)**|1.8001|0.3471|0.5882|0.7529|**0.8706**|
> |**BioMD-abs (AR-5)**|**1.7253**|**0.4912**|**0.6824**|**0.7853**|0.8588|
>
> **Notes**: *Metadynamics trajectory serves as the lower bound; metrics are computed across multiple repeating simulations.*

---

### Meta-Review · Area_Chair_EN82 · 2026-01-06

**Summary:**

This paper proposes BioMD, an all-atom generative model for biomolecular dynamics, with the goal of generating long-timescale trajectories efficiently.

**Reviewer Concerns:**

Initial reviews raised concerns about whether the method truly supports long-timescale dynamics, the adequacy of comparisons to strong generative baselines, and the level of physical validation. During the rebuttal, the authors made a substantial effort to address these points. In particular, they added new experiments on the ATLAS dataset, which go beyond the originally evaluated time horizon. These results suggest that the model can capture meaningful long-timescale conformational changes, including domain motions. The authors also introduced comparisons against stronger baselines (e.g., BioEmu and Str2Str) and reported improved performance on a majority of the evaluated metrics. One reviewer explicitly acknowledged these additions and increased their score accordingly.

Also, some concerns remain only partially resolved. The claimed advantages on long-timescale dynamics and robustness rely on a limited number of systems, and it is still unclear how broadly these results generalize. In addition, while the new comparisons and analyses are helpful, the presentation remains dense and some methodological details and assumptions could be clarified further.

Overall, the rebuttal significantly improves the paper and addresses several key reviewer concerns. Given the significance of the field and the methodological advances it presents, I am inclined to accept this manuscript.

**Reviewer Scores:**

Reviewer bQYZ has indicated an increase to a score of 8, while for the other reviewers I am inclined to keep their scores unchanged or consider a modest increase.

---

### Decision · Program_Chairs · 2026-01-26

Accept (Poster)